# Prediction of enzymatic pathways by integrative pathway mapping

Sara Calhoun[1†], Magdalena Korczynska[2†], Daniel J Wichelecki[3,4,5], Brian San Francisco[3], Suwen Zhao[2], Dmitry A Rodionov[6,7], Matthew W Vetting[8], Nawar F Al-Obaidi[8], Henry Lin[2], Matthew J O'Meara[2], David A Scott[6], John H Morris[9], Daniel Russel[1], Steven C Almo[8], Andrei L Osterman[6], John A Gerlt[3,4,5]*, Matthew P Jacobson[2]*, Brian K Shoichet[2]*, Andrej Sali[1,2,10]*

[1]Department of Bioengineering and Therapeutic Sciences, University of California, San Francisco, San Francisco, United States; [2]Department of Pharmaceutical Chemistry, University of California, San Francisco, San Francisco, United States; [3]Institute for Genomic Biology, University of Illinois, Urbana, United States; [4]Department of Biochemistry, University of Illinois, Urbana, United States; [5]Department of Chemistry, University of Illinois, Urbana, United States; [6]Sanford Burnham Prebys Medical Discovery Institute, La Jolla, United States; [7]A.A. Kharkevich Institute for Information Transmission Problems, Russian Academy of Sciences, Moscow, Russia; [8]Department of Biochemistry, Albert Einstein College of Medicine, New York, United States; [9]Resource for Biocomputing, Visualization and Informatics, Department of Pharmaceutical Chemistry, University of California, San Francisco, San Francisco, United States; [10]California Institute for Quantitative Biosciences, University of California, San Francisco, San Francisco, United States

*For correspondence:
j-gerlt@illinois.edu (JAG);
matt.jacobson@ucsf.edu (MPJ);
bshoichet@gmail.com (BKS);
Sali@Salilab.org (AS)

†These authors contributed equally to this work

**Abstract** The functions of most proteins are yet to be determined. The function of an enzyme is often defined by its interacting partners, including its substrate and product, and its role in larger metabolic networks. Here, we describe a computational method that predicts the functions of orphan enzymes by organizing them into a linear metabolic pathway. Given candidate enzyme and metabolite pathway members, this aim is achieved by finding those pathways that satisfy structural and network restraints implied by varied input information, including that from virtual screening, chemoinformatics, genomic context analysis, and ligand-binding experiments. We demonstrate this integrative pathway mapping method by predicting the L-gulonate catabolic pathway in *Haemophilus influenzae* Rd KW20. The prediction was subsequently validated experimentally by enzymology, crystallography, and metabolomics. Integrative pathway mapping by satisfaction of structural and network restraints is extensible to molecular networks in general and thus formally bridges the gap between structural biology and systems biology.
DOI: https://doi.org/10.7554/eLife.31097.001

## Introduction

### Problem and approach

The functions of most sequenced proteins have not been determined by experiment (*Gerlt et al., 2011*; *Jacobson et al., 2014*; *Schnoes et al., 2009*). They are also difficult to predict for enzymes with less than 60% sequence identity to characterized enzymes (*Radivojac et al., 2013*). The problem is much greater when seeking to predict the functions of entire metabolic pathways. Here, we propose a computational approach that outputs an enzymatic pathway and corresponding ligands,

given a set of potential enzymes and metabolites, using information derived by experiment and/or computational analyses (*Figure 1*). The approach benefits from two considerations. First, predicting an entire pathway may sometimes be easier than predicting individual enzymatic functions in isolation, because the product of one enzyme is the substrate for the next in the pathway. Thus, even when each enzyme's ligand assignment is ambiguous, the ligand assignments consistent with both enzymes may be more precise and accurate. Second, while it may be impossible to identify a pathway using information from any single method, there may be sufficient information provided by several methods. Our approach was inspired by the previous work using metabolite docking to multiple enzymes and substrate-binding proteins hypothesized to participate in the pathway (*Jacobson et al., 2014*; *Kalyanaraman and Jacobson, 2010*; *Macchiarulo et al., 2004*; *Zhao et al.,*

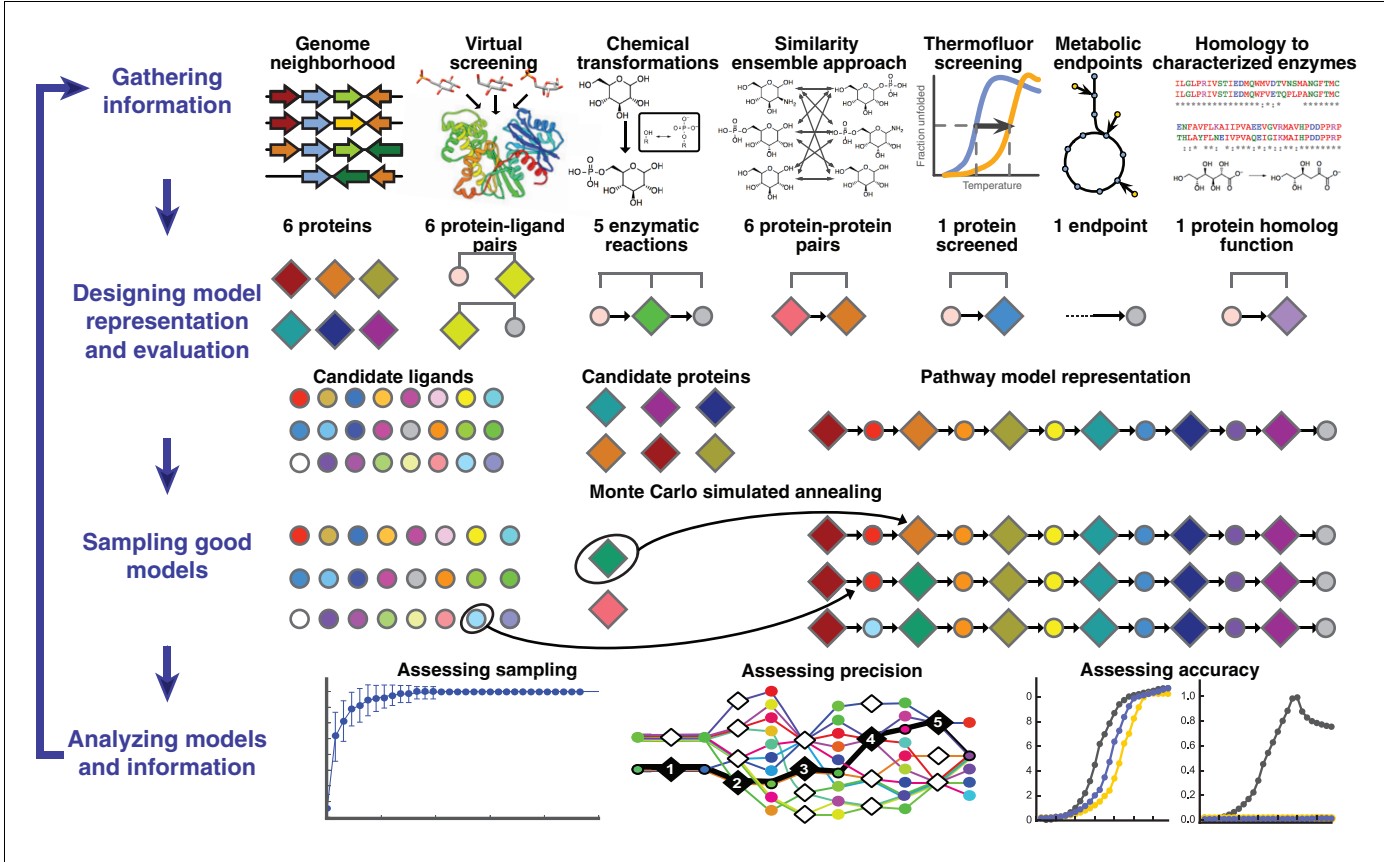

**Figure 1.** Overview of integrative pathway mapping method. The four stages of integrative modeling are: (1) Gathering information, (2) Designing model representation and evaluation, (3) Sampling good models, and (4) Analyzing models and information. (1) Here, the input information is gathered from seven different sources used to determine the candidate proteins, such as co-localization and conservation in the genome neighborhood, and the scoring restraints (docking scores from virtual screening, chemical transformations, ensemble similarity calculations of virtual screening hits from similarity ensemble approach, DSF screening hits, metabolic endpoints, and characterized chemical reactions). (2) A pathway model is represented as a graph composed of protein and ligand nodes. Proteins are depicted as diamonds and ligands are depicted as circles, with lines showing the node patterns evaluated by a given type of information. (3) The combinatorial optimization problem is solved by Monte Carlo simulated annealing sampling, consisting of randomly swapping nodes in and out of the pathway model and rearranging the edges between the nodes. (4) The final analysis stage involves assessing the sampling, precision, and accuracy of the models.

DOI: https://doi.org/10.7554/eLife.31097.002

The following figure supplements are available for figure 1:

**Figure supplement 1.** Workflow for preparing input data for the L-gulonate catabolic pathway prediction.

DOI: https://doi.org/10.7554/eLife.31097.003

**Figure supplement 2.** Pfam genome neighborhood network (GNN).

DOI: https://doi.org/10.7554/eLife.31097.004

**Figure supplement 3.** NetIMP cytoscape application for pathway model visualization.

DOI: https://doi.org/10.7554/eLife.31097.005

*2013*), integrative structure determination of large macromolecular assemblies (*Alber et al., 2007*; *Russel et al., 2012*), and comparative genomics approaches for metabolic reconstructions (*Markowitz et al., 2010*; *Karp et al., 2016*; *Osterman and Overbeek, 2003*; *Overbeek et al., 2005*; *Yamanishi et al., 2007*; *Ye et al., 2005*).

A pathway model is represented as a graph in which the enzymes, substrates, and products are nodes and the enzymatic reactions are edges (*Figure 1*). Input information, such as scores from experimental ligand screening, molecular docking screens, and chemical similarity, is encoded as 'network' restraints on the identity of the nodes and edges in the pathway; these restraints are combined into a scoring function. An ensemble of pathways consistent with the input information is computed by a Monte Carlo algorithm that samples well-scoring pathways over possible enzymes and metabolites. The resulting ensemble of good-scoring pathway models is assessed by its precision, its satisfaction of the input restraints, and ideally experimental observations not used in its construction. In addition to gauging the accuracy and precision of the models and the observations, this analysis can identify the most informative future experiments. Because the approach ranks alternative pathways using all available information, it in principle produces maximally accurate, precise, and complete pathway models given that information. The process of data gathering and modeling can iterate until a satisfactory model is obtained. We suggest that the four stages of integrative pathway mapping by satisfaction of network restraints mimic how human experts often derive and test pathway models.

## Results

The approach begins with a list of candidate proteins (here enzymes, binding proteins, and transporters) and a list of endogenous metabolites that are candidate substrates or products of these enzymes (*Figure 1*, *Figure 1—figure supplement 1*). The pathway members can be winnowed from the entire proteome by predicting functionally related proteins using information about the genome organization that is often available for bacterial pathways (*Zhao et al., 2014*). For example, for the gulonate pathway, we identified five metabolic enzymes that are conserved in the genome neighborhood of the TRAP transporter gene by constructing a genome neighborhood network (GNN) (*Figure 1—figure supplement 2*); the GNN approach has been demonstrated to accurately predict enzymes and transporters that function together in metabolic pathways based on conserved protein families in genome neighborhoods across different species (*Zhao et al., 2014*). The network restraints can then be inferred in multiple ways, exemplified by the following restraints in this study (*Figure 2A*). First, for each candidate, the libraries of endogenous metabolites are docked against either an experimentally determined structure if available or a comparative structure model (*Mysinger and Shoichet, 2010*). In the case of glycolysis, 2965 sugars in the KEGG database were screened against two crystal structures and eight comparative models for the 10 enzymes in this pathway (*Kalyanaraman and Jacobson, 2010*). Second, with the top 500 metabolites docked-and-ranked against each of the enzymes, the pathway enzymes may be linked by the similarity of their high-ranking docked ligands, here using the chemoinformatic Similarity Ensemble Approach (SEA) (*Keiser et al., 2007*; *Lin et al., 2013*); other related approaches can also be used (*Besnard et al., 2012*; *Gregori-Puigjané and Mestres, 2006*; *Mestres et al., 2006*; *Nidhi et al., 2006*; *Paolini et al., 2006*). The restraint can be informative because enzymes are often more likely to be pathway neighbors when their high-scoring docked ligands resemble each other. For instance, the top 500 metabolites of 3-phosphate dehydrogenase in the glycolysis pathway (as ranked by docking) are dominated by six chemotypes, while the phosphoglycerate kinase has three of these chemotypes overrepresented. This similarity is captured by the SEA E-value of $9.5 \ 10^{-63}$, suggesting that the observed level of similarity between the two predicted ligand lists is unlikely to have occurred by chance (*Figure 2—figure supplement 1C*). Thus, the two enzymes are linked by their related predicted metabolites. Third, consideration of the enzymatic reaction types assigned to the enzymes' superfamilies restrains the reactions in the pathway models. We require that the predicted metabolites can actually be substrates or products of an enzyme, given its reaction profile extracted from its protein family annotation. As an example, the glyceraldehyde 3-phosphate dehydrogenase is assigned the reactions that can convert an aldehyde to a phosphate and *vice versa* (*Supplementary file 6*). Finally, all available experimental screening hits, substrate specificities from homology, constraints on the pathway endpoints, and other information can also be considered.

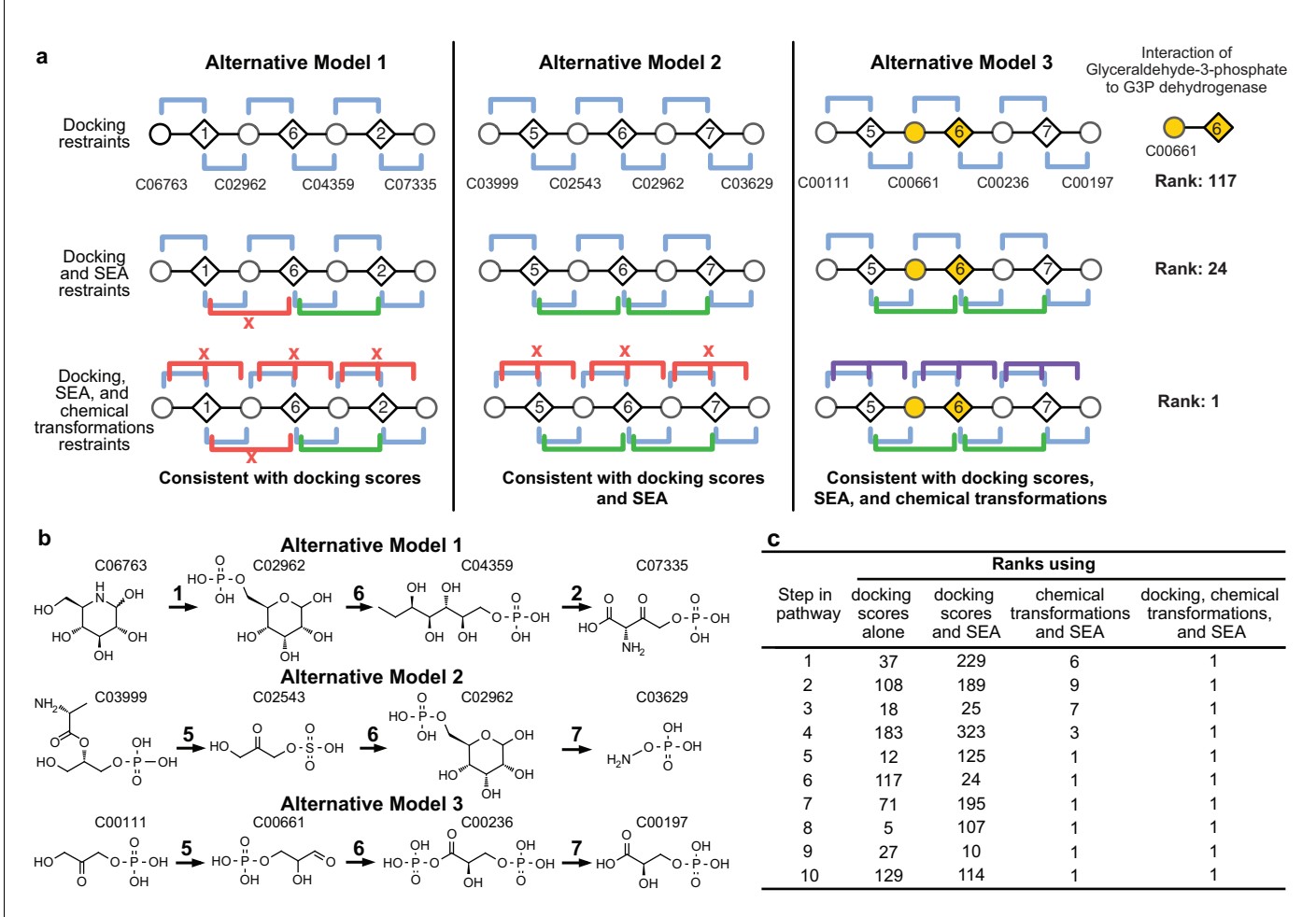

**Figure 2.** Representation of alternative models obtained based on consistency with input information provided for the glycolysis benchmark pathway. (A) Example of three alternative models evaluated using different types of restraints based on modeling of the glycolysis pathway with a subset of pathways shown. The restraints on node patterns are shown using colored lines (blue – docking restraints, green – SEA restraints, purple – chemical transformation restraints, red – restraints with unfavorable scores). Metabolites are labeled by KEGG ID and enzymes are labeled by step in glycolysis pathway. On the left, alternate model one is consistent with docking scores, but not with all SEA scores and chemical transformations. In the middle, alternate model two is consistent with the docking scores and SEA scores, but not with chemical transformations. On the right, alternate model three is consistent with docking scores, SEA scores, and chemical transformations, thus increasing the rank of the correct enzyme-substrate pairings. (B) Alternative models shown with chemical structures. (C), Ranks of correct substrate for the corresponding enzyme at each step in the glycolysis benchmark case. 1 – glucokinase, 2 – phosphoglucose isomerase, 3 – phosphofructokinase, 4 – fructose bisphosphate aldolase, 5 – triosephosphate isomerase, 6 – glyceraldehyde 3-phosphate dehydrogenase, 7 – phosphoglycerate 8 – phosphoglycerate mutase 9 – enolase and 10 – pyruvate kinase.
DOI: https://doi.org/10.7554/eLife.31097.006

The following figure supplement is available for figure 2:

**Figure supplement 1.** Benchmark assessment for decoy and dummy enzymes.
DOI: https://doi.org/10.7554/eLife.31097.007

Each of these considerations is added to a scoring function that ranks alternative pathway models by assessing their consistency with the available information (*Figure 2B*). Thus, pathway models are preferred when they contain (i) good-scoring metabolite-enzyme pairs, (ii) pairs of neighboring enzymes that share chemotypes, (iii) pairs of neighboring enzymes catalyzing chemical reactions that allow the product of an upstream enzyme to be a substrate of the downstream enzyme, *etc*. This integrative approach does not require that each type of restraint be available for each protein and metabolite, nor that each restraint is accurate or precise; it only requires that the scoring function consisting of all restraints is sufficiently accurate and precise.

## Benchmarking

The method was tested by retrospectively ordering enzymes and identifying their substrates in three well characterized pathways: glycolysis (10 enzymes) (*Kalyanaraman and Jacobson, 2010*), cytidine monophosphate 3-deoxy-D-*manno*-octulosonate 8-phosphate (CMP KDO-8P) biosynthesis (four enzymes), and serine biosynthesis (five enzymes) (*Supplementary files 1* and *2*). Docking screens of several thousand metabolites against comparative models of the enzymes, chemical transformation annotations of the enzymes, and the chemoinformatic SEA analysis were the input information for mapping each pathway. Because the functions of these enzymes have been characterized, homology-based annotations were not included as restraints for the purposes of the retrospective benchmark. The method successfully identified the substrates and products and correctly ordered all pathway components in the top-scoring models (*Figure 2*; *Supplementary files 1* and *2*). The method performed well even when the number and identity of pathway enzymes were unspecified (*Figure 3AB*) or when the candidate enzymes set was incomplete (*Figure 2—figure supplement 1D*).

The accuracy of the predicted benchmark pathways is not limited by the lack of sampling (*Figure 3—figure supplement 1*), but rather by the input information (*Figure 2A*). Thus, the integration of multiple types of information improves the accuracy and precision of pathway prediction (*Figure 2BC*). For example, the correct substrate of the dehydrogenase in the glycolysis pathway, glyceraldehyde 3-phosphate (G3P), is predicted to be most consistent with all the input information. Although by docking alone, the rank of G3P is only 117 out of the 2965 metabolites docked, the additional restraints from SEA and chemical transformations lead to the overall top ranking of G3P (*Figure 2A*).

Several caveats bear mentioning. Both thorough sampling and accurate scoring become more difficult when the number of possible pathways increases (which in turn arises from a large set of candidate enzymes and metabolites), when some enzymes or metabolites are not in the input set, when the pathway is long, or its length is unknown. Here, only linear pathways are sampled; thus, non-linear pathways, including cyclic pathways, are not modeled. The preparation of input information requires manual processing. Although docking, chemoinformatics, comparative modeling, chemical transformations, and differential scanning fluorimetry (DSF) screening information may be collected in an automated way, the quality of information often benefits from expert choices. For example, comparative model building can be especially time consuming when low sequence similarity structures are available for target building, and docking may require expert intervention when parameterization of cofactors is necessary for correctly defining the binding site. Nevertheless, we emphasize that once the input information is provided, its conversion into the predicted pathway is automated and does not require human intervention. Finally, docking against modeled structures will sometimes fail, even with the added advantages of insisting on consistency in docking hit lists. Some of these pitfalls can be detected through testing the thoroughness of sampling (*Figure 3—figure supplement 1*), statistical bootstrapping and jack-knifing tests (*Efron, 1981*), and by direct experimental testing of predictions (*Figure 4*). The method becomes more robust when the pathway start and end are defined. More generally, failures can also be reduced by introducing restraints or constraints that limit the size of the input enzyme and metabolite sets, by improving the accuracy of the scoring function, by limiting the sampling, or by further filtering the set of good-scoring solutions.

## Prospective prediction of the L-gulonate catabolic pathway in *Haemophilus influenzae* Rd KW20

L-gulonate and D-mannonate were identified as potential ligands of the TRAP solute binding protein (SBP) from *H. influenzae* (*Supplementary file 3*), using DSF screening of a library of 189 compounds (*Vetting et al., 2015*). The *Hi*GulPQM TRAP transporter consists of three subunits, including the periplasmic SBP *Hi*GulP, and two membrane components *Hi*GulQ and *Hi*GulM. SBPs recognize substrates to be imported into the cytosol by the transporter. Because these sugars are not involved in central carbon metabolism, the observation suggested an uncharacterized pathway that converts L-gulonate or D-mannonate into substrates for central carbon metabolism. While this pathway had been proposed based on the DSF screening hits and genome neighborhood analysis, we sought to predict it using integrative pathway mapping, based on the following information (*Figure 1*, *Figure 1—figure supplement 1*).

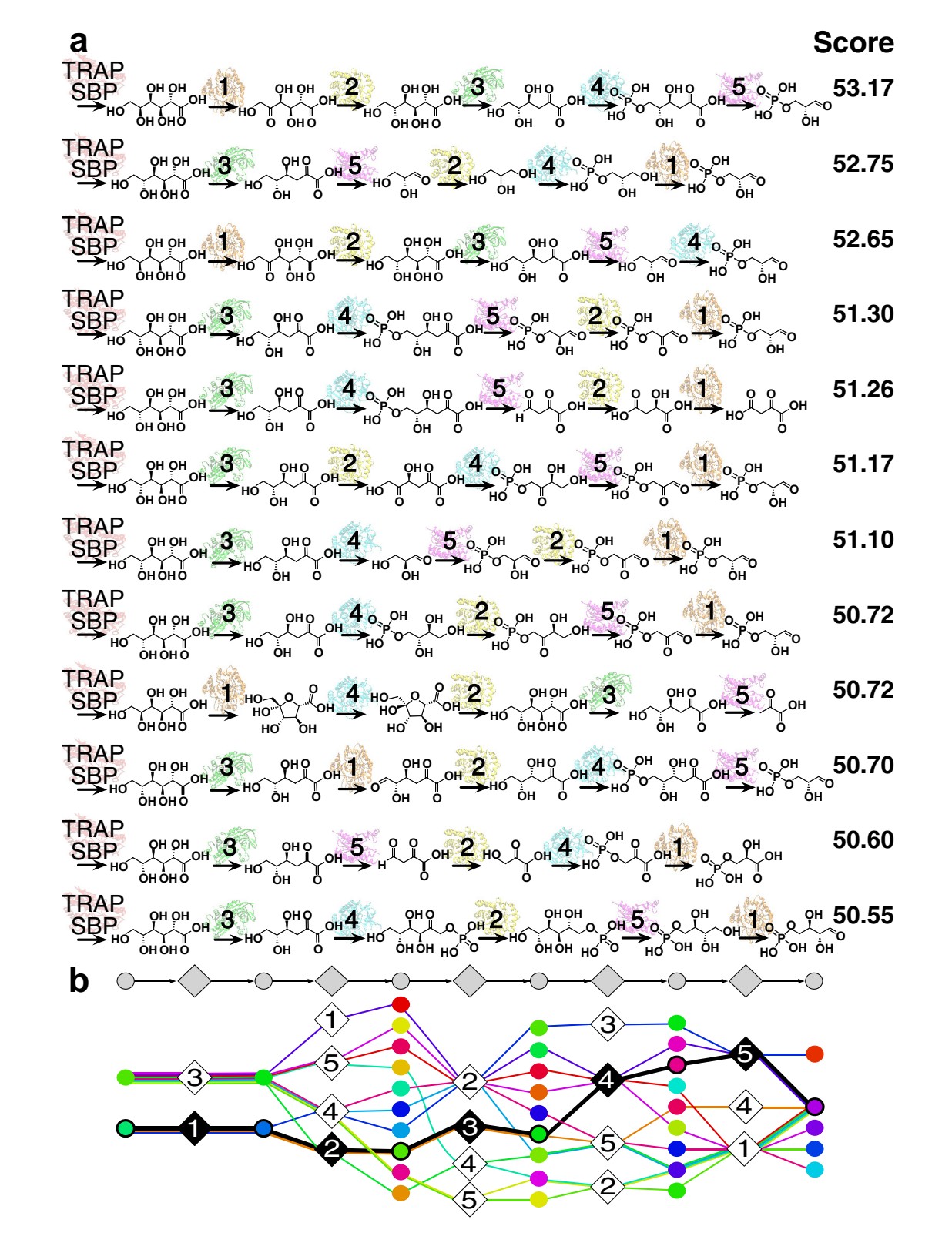

**Figure 3.** 12 representative predictions of the L-gulonate TRAP-SBP catabolic pathway. (A) 12 representative pathway models of TRAP SBP pathway predictions ordered by score, starting from the top with the best-scored prediction. The scores of the representative pathways are listed to the right of the corresponding pathway. Pathway enzymes are labeled by numbers as follows: 1 – *Hi*GulD, 2 – *Hi*UxuB, 3 – *Hi*UxuA, 4 – *Hi*KdgK, 5 – *Hi*KdgA. (B) Graphical representation of an ensemble of representative pathway models. The predicted components in the ensemble of pathway models at each

*Figure 3 continued on next page*

*Figure 3 continued*

position are vertically aligned to the corresponding position in the gray pathway on the top. Ligand components are shown as circle nodes with the color corresponding to the ligand identity. Chemical structures are shown in *Figure 3—figure supplement 2*. Pathway enzymes are shown as diamond nodes with the same numbering as above. Edges are colored by individual pathway model prediction. The validated prediction is shown by black edges, enzyme nodes are colored black, and substrate/product nodes are outlined in black.

DOI: https://doi.org/10.7554/eLife.31097.008

The following figure supplements are available for figure 3:

**Figure supplement 1.** Sampling convergence test.
DOI: https://doi.org/10.7554/eLife.31097.009
**Figure supplement 2.** Chemical structures for top scoring pathway model predictions.
DOI: https://doi.org/10.7554/eLife.31097.010

First, the position of TRAP SBP was fixed at the pathway start and its ligand was constrained to be L-gulonate or D-mannonate; this positioning is reasonable given the TRAP's role as a transporter. Second, five possible pathway enzymes (a dehydratase, two dehydrogenases, a kinase, and an aldolase) were identified from the genome neighborhood around the TRAP-solute-binding protein (Uniprot ID P71336) (*Figure 1—figure supplement 2A–C*, *Supplementary file 2*). Third, 3650 out of the 14,212 metabolites in KEGG (*Kanehisa et al., 2016*) were identified as the smallest unique ligand set of substrates or products for these enzymes, based on the top scoring docking hits, which were optimized by chemical transformations and chemical similarity (*Supplementary file 2*). Fourth, the pathway was constrained to end in a metabolite of central metabolism (*Supplementary file 4*). Fifth, the dehydratase (Uniprot ID P44488) was hypothesized to be a D-mannonate dehydratase because of high sequence similarity to a characterized D-mannonate dehydratase, UxuA, in other organisms (73% sequence identity to UxuA in *E. coli*) (*Dreyer, 1987*). Finally, as in the benchmarking, the scoring function used docking, SEA E-values, and chemical transformations inferred from annotations in Pfam (*Finn et al., 2016*).

The sampling algorithm found 154 unique high-scoring pathways, which clustered into 12 groups (*Figure 3AB*). The best-scoring pathway, starting from L-gulonate as the ligand of the TRAP SBP, begins with its oxidation to D-fructuronate by the first dehydrogenase (*Hi*GulD) as the first catalytic step (*Figure 4A*). Next, reduction of D-fructuronate by the second dehydrogenase (*Hi*UxuB) produces D-mannonate; its dehydration by the dehydratase (*Hi*UxuA) produces 2-keto-3-deoxy-D-gluconate. The last few steps in the pathway model are part of a conserved Entner-Dourdoroff pathway, ending with glyceraldehyde 3-phosphate and pyruvate, known members of central metabolism. Even when individual restraints are excluded from the input information, the best-scoring pathway falls within the top-scoring pathway models. For example, the same pathway has the highest score if either the starting or end point is not restrained. If information about both starting and end points is excluded, this pathway model drops to 15th in score.

## Experimental testing of the L-gulonate catabolic pathway

The pathway model was tested experimentally in five independent ways, including by enzyme activity, X-ray crystallography, fitness growth assays of the deletion mutants, transcript analyses, and isotopic metabolic labeling (*Figure 4A–E*).

First, all enzymes had $k_{cat}/K_M$ values larger than $10^3$ $M^{-1}s^{-1}$ for their predicted substrates (*Figure 4B*). Initially, *Hi*GulD had negligible activity with its putative substrate L-gulonate. However, the pathway prediction was deemed to be of sufficient quality to encourage optimization of enzyme purification, ultimately producing an active enzyme with a $k_{cat}/K_M$ value of $10^4$ $M^{-1}s^{-1}$ for L-gulonate. All enzymes exhibit micromolar $K_M$ values, except for *Hi*UxuA (potentially reflecting high D-mannonate concentrations in the cytosol). Second, the model is supported by the crystallographic structure of the complex between the TRAP SBP protein and L-gulonate (*Vetting et al., 2015*) (*Figure 4C*). Third, knockouts (KOs) of Δ*Hi*GulP SBP and Δ*Hi*GulD dehydrogenase were constructed. Δ*Hi*GulP and Δ*Hi*GulD KO strains retain the ability to grow on glucose (*Figure 4D*, left), while they do not grow on L-gulonate (*Figure 4D*, right). Fourth, all predicted pathway encoding genes, including *Hi*GulPQM transporter, *Hi*GulD, *Hi*UxuA, *Hi*UxuB, *Hi*KdgK, and *Hi*KdgA, are upregulated when *H. influenzae* is grown on L-gulonate or D-mannonate as the sole carbon source (*Figure 4E*). Fifth, when *H. influenzae* was incubated with U-$^{13}$C-L-gulonate during the early exponential phase, even

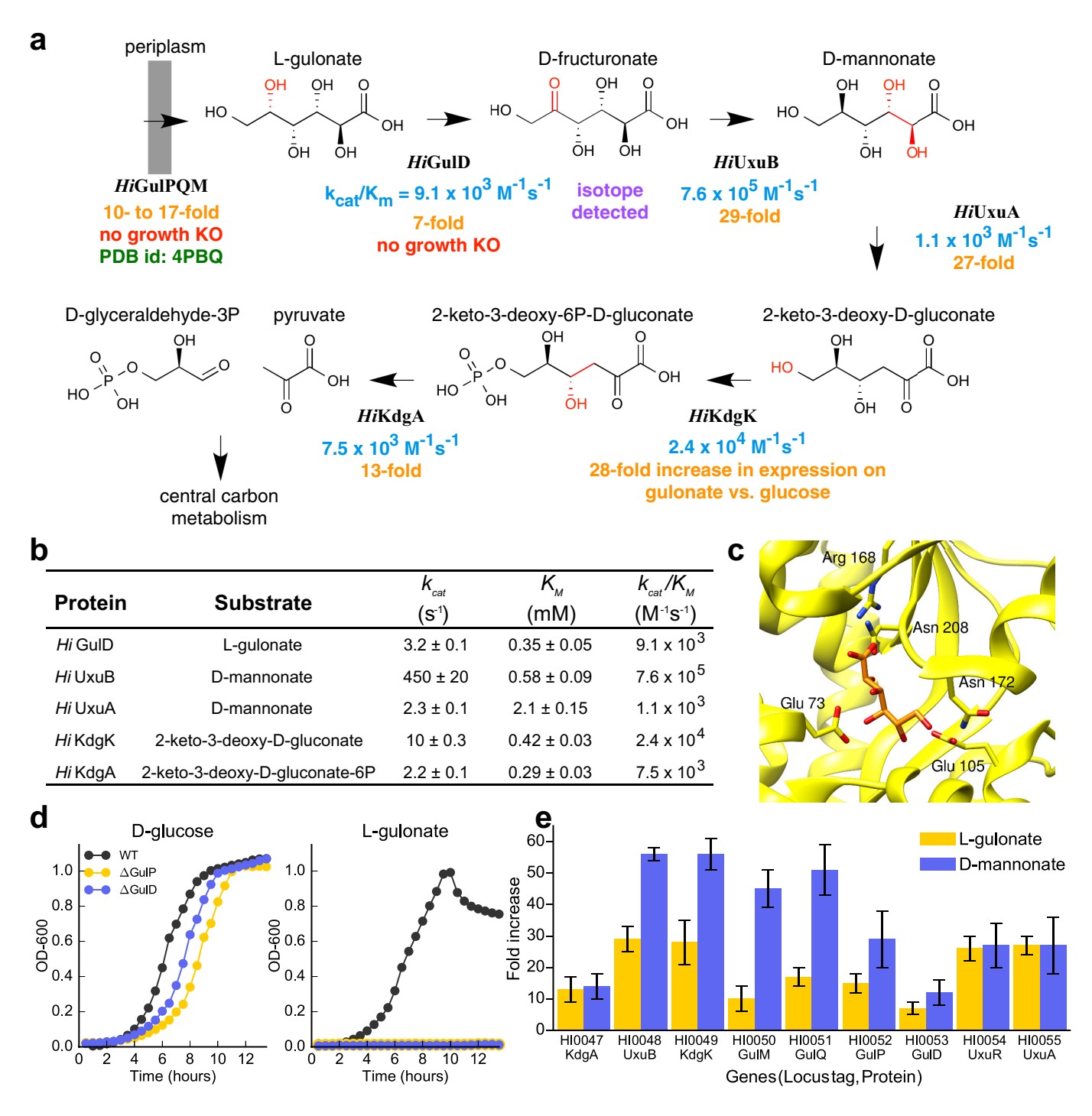

**Figure 4.** Catabolic pathway of *H. influenzae* Rd KW20. (**A**) The best-scoring pathway identified using the integrative mapping approach is annotated with experimental evidence: enzyme activity (blue), fitness growth determinants (red), transcript analyses on L-gulonate media (orange), atomic structure (green), and isotopic metabolic labeling (purple). The pathway demonstrates L-gulonate degradation into glyceraldehyde 3-phosphate and pyruvate. Bonds undergoing changes in the subsequent steps are colored in red. (**B**) Kinetics of pathway enzymes on predicted substrates. (**C**) Crystal structure of L-gulonate bound to SBP TRAP (PDB ID: 4PBQ). (**D**) Knockout growth assays of *H. influenzae* strains, ΔGulP (gulonate transporter periplasmic subunit) and ΔGulD (L-gulonate dehydrogenase), when grown on D-glucose vs. L-gulonate as a sole carbon source. (**E**) Fold change in expression for each gene when grown on the indicated carbon source, relative to growth on glucose. Error bars indicate one standard deviation for three biological replicates.

DOI: https://doi.org/10.7554/eLife.31097.011

*Figure 4 continued on next page*

*Figure 4 continued*

The following figure supplements are available for figure 4:

**Figure supplement 1.** Isotopic labeling of L-gulonate as sole carbon source.

DOI: https://doi.org/10.7554/eLife.31097.012

**Figure supplement 2.** Comparative genomic reconstruction of L-gulonate and related uronic acid catabolic pathways and regulons in gammaproteobacteria.

DOI: https://doi.org/10.7554/eLife.31097.013

for one minute, substantial labeling of central carbon metabolites was observed, indicating rapid cellular uptake and metabolism of L-gulonate (*Figure 4—figure supplement 1A*). In addition, time-dependent labeling of D-fructuronate was observed (*Figure 4—figure supplement 1B*), further supporting the first predicted step in the L-gulonate catabolic pathway (*Figure 4—figure supplement 1C*). Finally, identification of this pathway in *H. influenzae* allowed us to reconstruct the L-gulonate and hexuronate pathways in related bacteria, mapping their conservation and variation to better understand the evolution and function of the pathway (*Figure 4—figure supplement 2*).

## Discussion

A number of methods have been developed to predict metabolic enzymes and pathways (*Jacobson et al., 2014*; *Planes and Beasley, 2008*). The most common method assigns enzyme function from its sequence similarity to a characterized enzyme (*Lee et al., 2007*), sometimes allowing genome-scale metabolic reconstructions (*Karp et al., 2016*; *Bordbar et al., 2014*). However, similarity-based approaches often fail when the sequence identity drops below 60% (*Radivojac et al., 2013*) or when distant homologs are functionally divergent. Virtual screening used for integrative pathway mapping, even against comparative models, can predict substrates more accurately than homology-based transfer (*Fan et al., 2009*). In other cases, functional linking based on omics data, such as gene clusters, phylogenetic profiles, and gene expression profiles, can also guide functional prediction (*Osterman and Overbeek, 2003*; *Overbeek et al., 1999*). Methods that integrate sequence similarity and functional linking can improve predictions (*Plata et al., 2012*), as can approaches that incorporate modeling of metabolic flux with genomics-based metabolic reconstruction to identify missing enzymes (*Karp et al., 2016*; *Bordbar et al., 2014*; *Monk et al., 2014*). Several studies have combined structural information with metabolic reconstructions for genome-scale analysis (*Zhang et al., 2009*; *Brunk et al., 2016*; *Chang et al., 2010*). Still, most of the methods are limited by the biochemical knowledge available and the reactions that are mapped. Studies that deorphanize enzyme function (*Irwin et al., 2005*; *Korczynska et al., 2014*) or annotate new pathways (*Zhao et al., 2013*) will enhance the accuracy and applicability of these computational methods (*Bordbar et al., 2014*) as well as our integrative method. However, a key strength of our integrative approach is its ability to predict pathways that contain previously unknown biochemical reactions, and to assemble pathways de novo from simple and often newly predicted enzyme activities.

Integrative pathway mapping provides a flexible and general approach to functional annotation and pathway modeling. Because it generalizes functional annotation into the sampling of pathways consistent with any available input information, it can use more information than alternative methods and thus, at least in principle, produces more accurate, precise, and complete answers. For example, while there are numerous methods for predicting functions by combining information (*Yamanishi et al., 2007*; *Ye et al., 2005*; *Plata et al., 2012*; *Green and Karp, 2007*; *Kharchenko et al., 2006*; *Smith et al., 2012*; *Zhu et al., 2012*), the generality and flexibility of integrative pathway mapping allows us to combine structural information with other types of data in a most straightforward manner. If not all bacterial pathways have enough information from the genome context to infer the pathway enzymes, many do. Moreover, no single type of input information is essential, provided sufficient information is available from other sources. For example, potential pathway members in prokaryotes could also be obtained from regulon analysis based on predicting conserved binding sites for transcriptional regulators (*Ravcheev et al., 2013*; *Rodionova et al., 2013*). Other approaches for identifying candidate pathway members are especially needed for eukaryotes, because the relationship between genome neighborhood and pathway

membership is significantly weaker in eukaryotes than in prokaryotes. For some pathways in eukaryotes, consideration of homology and biochemical function as well as direct experimental evidence, such as spatial co-localization by proteomics or chemical cross-linking, could be used to identify a set of potential protein members for pathway mapping. Thus, the integrative approach is at least in principle not limited to prokaryotic pathways. If docking struggles to prioritize the right substrates as top ranking hits, it often ranks the right ones well (*Korczynska et al., 2014*; *Hall et al., 2010*; *Hermann et al., 2007*); insisting that the product of one step feed into the next provides a surprisingly useful criterion not only for pathway membership and ordering, but also for re-prioritizing the correct substrate from the docking candidates. The integrative approach strengthens what would ordinarily be approximate answers by insisting on maximal possible consistency across the enzymes and across different types of information. Because of the generality of integrative pathway mapping, new sources of information can be incorporated, including knockout screens and known metabolic capabilities. While not all types of input for integrative pathway mapping can be obtained automatically (e.g. docking, experimental measurements), the mapping itself is entirely automated. With further development, the framework may be applicable on a larger scale, approaching complete genomes, but mapping topologies of networks will be more demanding as it will require more input information and larger computation.

# Materials and methods

## Key resources table

| Reagent type (species) or resource | Designation | Source or reference | Identifiers | Additional information |
|---|---|---|---|---|
| Gene (*Haemophilus influenzae*) | UxuA | This paper, pNYCOMPSC-tagless HiUxuA vector | Uniprot:P44488 | See *Supplementary file 7*. cloned into the C-terminal TEV cleavable 10x-Histag containing vector pNYCOMPS-LIC-TH10-ccdB (C-term) such that the tag is out of frame |
| Gene (*H. influenzae*) | GulD | This paper, pNYCOMPSC-tagless HiGulD vector | Uniprot:Q57517 | See *Supplementary file 7*. cloned into the C-terminal TEV cleavable 10x-Histag containing vector pNYCOMPS-LIC-TH10-ccdB (C-term) such that the tag is out of frame |
| Gene (*H. influenzae*) | KdgK | This paper, HiKdgK-pSGC-His vector | Uniprot:P44482 | See *Supplementary file 7*. cloned into the N-terminal TEV cleavable 6x-Histag containing vector pNIC28-Bsa4 |
| Gene (*H. influenzae*) | UxuB | This paper, HiUxuB-pSGC-His vector | Uniprot:P44481 | See *Supplementary file 7*. cloned into the N-terminal TEV cleavable 6x-Histag containing vector pNIC28-Bsa4 |
| Gene (*H. influenzae*) | KdgA | This paper, HiKdgA-pSGC-His vector | Uniprot:P44480 | See *Supplementary file 7*. cloned into the N-terminal TEV cleavable 6x-Histag containing vector pNIC28-Bsa4 |
| Gene (*H. influenzae*) | GulP | This paper | Uniprot:P71336 | See *Supplementary files 8* and *9* |
| Gene (*H. influenzae*) | GulQ | This paper | Uniprot:P44484 | See *Supplementary files 8* and *9* |
| Gene (*H. influenzae*) | GulM | This paper | Uniprot:P44483 | See *Supplementary files 8* and *9* |
| Gene (*H. influenzae*) | UxuR | This paper | Uniprot:P44487 | See *Supplementary files 8* and *9* |
| Oligonucleotide (*H. influenzae*) | UxuA, UxuR, GulD, GulP, GulQ, GulM, KdgK, UxuB, KdgA, Hflu | This paper | | See *Supplementary file 8*. qRT-PCR oligonucleotide sequences used for gene expression profiling |
| Strain, strain background (*H. influenzae* Rd KW20) | H. flu | https://www.atcc.org | ATCC 51907 | *Supplementary file 9*. Genetic deletion mutants of the putative L-gulonate catabolism pathway in *H. influenzae* Rd KW20 |

*Continued on next page*

*Continued*

| Reagent type (species) or resource | Designation | Source or reference | Identifiers | Additional information |
|---|---|---|---|---|
| Genetic reagent (*H. influenzae*) | ΔGulP | This paper | | ***Supplementary file 8**. Genetic deletion mutants of the putative L-gulonate catabolism pathway in *H. influenzae* Rd KW20 |
| Genetic reagent (*H. influenzae*) | ΔGulD | This paper | | ***Supplementary file 8**. Genetic deletion mutants of the putative L-gulonate catabolism pathway in *H. influenzae* Rd KW20 |
| Transfected construct (*E. coli* BL21 (DE3) | BL21 (DE3) *E. coli* containing the pRIL plasmid | Stratagene | | Growth media contain 25 μg/mL Kanamycin or 100 μg/mL Carbomycin and 34 μg/mL Chloramphenicol |
| Commercial assay or kit | RNAprotect Bacteria Reagent | Qiagen | Cat No./ID: 76506 | |
| Commercial assay or kit | RNeasy Mini Kit | Qiagen | Cat No./ID: 74104 | |
| Commercial assay or kit | ProtoScript First Strand cDNA Synthesis Kit | New England BioLabs | Cat No./ID: E6300S | |
| Chemical compound, drug | 2-keto-3-deoxy-D-gluconate | Enzymatically synthesized | CAS: 17510-99-5 | Enzymatic synthesis by D-mannonate dehydratase (Uniprot ID B0T0B1). Verified via 1H-NMR |
| Chemical compound, drug | 2-keto-3-deoxy-D-gluconate-6-phosphate | Enzymatically synthesized | CAS: 884312-23-6 | Enzymatic synthesis by D-mannonate dehydratase (Uniprot ID B0T0B1) and 1 μM 2-keto-3-deoxy-D-gluconate kinase (Uniprot ID A4XF21). Verified via 1H-NMR |
| Software, algorithm | Integrative Pathway Mapping | This paper | https://github.com/salilab/pathway_mapping | The source code for the IMP program, benchmark, input scripts files, and output files for the benchmark and the gulonate pathway calculations are available here(50) |
| Software, algorithm | IMP program | Russel D, *et al*, Putting the pieces together: integrative structure determination of macromolecular assemblies. PLoS Biology. 10(1):e1001244, 2012 | http://integrativemodeling.org | Integrative modeling |
| Software, algorithm | MODELLER | B. Webb, A. Sali. Comparative Protein Structure Modeling Using Modeller. Current Protocols in Bioinformatics, John Wiley & Sons, Inc., 5.6.1-5.6.32, 2014. | https://salilab.org/modeller/ | Comparative modeling |
| Software, algorithm | DOCK3.6 | Mysinger MM, Shoichet BK. Rapid context-dependent ligand desolvation in molecular docking. J Chem Inf Model. 50(9):1561-73, 2010. | http://dock.compbio.ucsf.edu/ | Docking |
| Software, algorithm | Automated version DOCK3.6 | Irwin JJ, *et al*. Automated Docking Screens: A Feasibility Study. J. Med. Chem. 52(18)5712–5720, 2009. | http://blaster.docking.org/ | Docking |
| Software, algorithm | Similarity Ensemble Approach (SEA) | Keiser MJ, *et al*. Relating protein pharmacology by ligand chemistry. Nat Biotechnol. 25(2): 197-206, 2007. | http://sea.bkslab.org/ | SEA chemo-informatic calculations |
| Software, algorithm | OpenEye Scientific Software | OpenEye Scientific Software I. OEChem. 2.0.2 ed2014. | https://www.eyesopen.com/ | In silico chemical transformations |
| Software, algorithm | RDKit | Landrum G. RDKit: Open-source cheminformatics. Release_2016.03.1 ed2016 | http://www.rdkit.org/ | Chemical similarity calculations |

*Continued on next page*

*Continued*

| Reagent type (species) or resource | Designation | Source or reference | Identifiers | Additional information |
|---|---|---|---|---|
| Software, algorithm | EFI-EST | Gerlt JA, *et al.* Enzyme Function Initiative-Enzyme Similarity Tool (EFI-EST): A web tool for generating protein sequence similarity networks. Biochim. Biophys. Acta. 1854(8):1019-1037, 2015. | http://efi.igb.illinois.edu/efi-est/index.php | Genome neighborhood networks |
| Software, algorithm | Pythoscape v1.0 | Barber AE, Babbitt PC. Pythoscape: a framework for generation of large protein similarity networks. Bioinformatics. 28(21):2845-2846, 2012. | http://www.rbvi.ucsf.edu/trac/Pythoscape | Sequence similarity networks |
| Software, algorithm | Cytoscape v3.4 | Shannon P, *et al.* Cytoscape: a software environment for integrated models of biomolecular interaction networks. Genome Res. 13(11):2498-504, 2003. | http://www.cytoscape.org/ | Network visualization |

## Computational methods

### Integrative pathway mapping

The method computes all linear pathway models consistent with the input information, in a four-stage process (*Figure 1*). First, input information has to be collected from computational and/or experimental sources. Here, three established and convenient computational methods (i.e. molecular docking, Similarity Ensemble Approach, and chemical transformation analysis) were selected to illustrate the idea of integrative pathway mapping and to benchmark it on three known pathways. In principle, subsets of input information can be missing. Moreover, additional types of information can be added, hopefully improving the accuracy, precision, and applicability of the approach, as illustrated by the gulonate pathway prediction that also depends on DSF data, pathway anchor points, and protein homology considerations. Second, each data point is converted into a pathway restraint via a Z-score. The score of a pathway model is then a sum of these Z-scores. Third, the good scoring pathways are found by Monte Carlo sampling of pathways consisting of input enzymes and metabolites. Finally, the good scoring pathways are analyzed. Next, we describe the four stages of integrative pathway mapping in turn, using the L-gulonate catabolic pathway as an example (*Figure 1*).

### Stage 1: Gathering information

Information for the pathway mapping cases comes from the following sources: high-throughput DSF screening, genome context, structure-based docking screens, chemical transformations based on Pfam classification (*Finn et al., 2016*), and knowledge of central metabolism. With this information in hand, we use it to design representation, scoring, and sampling, which determine the output models.

### Stage 2: Designing pathway model representation and evaluation

For pathways of unknown length, we model pathways of each possible length independently, and then select an optimal combination of pathway length and score. The pathway model is represented as a linear graph, in which the molecular components are represented by nodes and the interactions are represented by edges. In the specific case of a metabolic pathway, the two classes of molecular components are the metabolites (substrates and products) and the proteins, which are binding proteins, transporters, or enzymes converting substrates to products. In addition, we allow for the inclusion of a dummy node representing an unknown and uncharacterized protein in the pathway. The sampling space of the models is constrained by the candidate enzyme and metabolite node identities that are given as input, as well as the linearity and length of the pathway.

A sequence similarity network (SSN) and genome neighborhood network (GNN) were constructed using the EFI-EST webserver (*Gerlt et al., 2015*) and Pythoscape v1.0 software (*Barber and*

*Babbitt, 2012*) for an anchor protein, TRAP SBPs (Uniprot ID P71336 and Uniprot ID A7JQX0), to provide candidate pathway members (*Zhao et al., 2014*; *Gerlt et al., 2015*). The network stringency for computing iso-functional clusters was set to an E-value cutoff of $10^{-120}$, corresponding to a median sequence identity between proteins of ~60% (*Vetting et al., 2015*). At this stringency, the majority of experimentally annotated TRAP SBPs are assigned to isofunctional clusters in the SSN. The full GNN was clustered based on Pfam designation into individual neighborhood nodes in the genome neighborhood of cluster 223, which included the TRAP transporter anchor protein (*Figure 1—figure supplement 2*). Analysis of the GNN identified five enzyme families as candidate pathway members, including two dehydrogenases, one sugar dehydratase, one carbohydrate kinase, and one aldolase. The genes associated with these families in *H. influenzae* are colocalized in the genome with the TRAP SBP gene. This step can be substituted or supplemented by any other method that identifies candidate genes, including but not limited to: (1) colocalization of genes providing operon/metabolic context for prokaryotic proteins (*Overbeek et al., 1999*), (2) coexpression measured through chip-based and RNA-seq technologies (*Wang et al., 2009*), (3) coregulation predicted by upstream DNA motifs (*Pilpel et al., 2001*; *Rodionov, 2007*), (4) protein-protein interaction studies (*Bork et al., 2004*; *Meier et al., 2013*), (5) protein fusion events (*Enright et al., 1999*; *Marcotte et al., 1999*), and (6) phylogenetic profiles across different genomes (*Pellegrini et al., 1999*).

To obtain the smallest candidate subset of KEGG that contains all metabolites needed to predict a pathway, we considered only the metabolites with good virtual screening scores against any of the candidate proteins as well as metabolites that can be derived from the virtual screening hits by applying chemical transformations related to the known activities of enzymes in the relevant superfamilies. Therefore, the top 1000-scoring metabolites from each docking screen are added to a single list of metabolites. Chemical transformations performed by each predicted enzyme are applied on the top-scoring metabolites using OEChem Tools (*OpenEye Scientific Software I, 2014*) excluding metabolites with no matches to the substrate motifs. Products of these reactions are compared by RDKit Morgan fingerprints (*Landrum, 2016*; *Rogers and Hahn, 2010*) to the metabolites from the KEGG LIGAND database (*Kanehisa et al., 2016*; *Kanehisa and Goto, 2000*). KEGG metabolites that have a Tanimoto coefficient above 0.75 to the products are added to the list of metabolites. This final list of metabolites contains 3650 unique ligands that are considered as the sampling space for candidate metabolite nodes.

## Scoring pathway models

Information about the pathway is encoded as pathway restraints that are summed into a scoring function. For example, a candidate edge between a given enzyme and metabolite is restrained by a virtual screening score for the pair. In an attempt to 'weigh' each piece of information optimally, each term in the scoring function is expressed as a Z-score. Our scoring function can in principle benefit from all available information, even when some information is not available for every enzyme or ligand. In such cases, the corresponding terms are simply omitted from the scoring function. A dummy node in a model contributes no score, except towards the chemical transformation term. Thus, the scoring function ($Z_{Pathway}$) for ranking alternative L-gulonate pathways is a sum of Z-scores for each type of restraint, including virtual screening ($Z_{VS}$), chemical transformations ($Z_{CT}$), SEA analysis ($Z_{SEA}$), known pathway boundaries ($Z_{CM}$), high-throughput screening ($Z_{HTS}$), and homology to characterized enzymes ($Z_{HS}$):

$$Z_{Pathway} = Z_{VS} + Z_{CT} + Z_{SEA} + Z_{CM} + Z_{HTS} + Z_{HS}$$

Next, we define these specific pathway restraints.

## Molecular docking screens

Favorable binding interactions predicted by docking can illuminate the identity of a ligand-protein pair. Pathway models with ligand-protein pairs that have favorable docking scores are more likely to be correct than those that have unfavorable docking scores. For each candidate pathway protein, a crystal structure or homology model, generated by MODELLER (*Sali and Blundell, 1993*; *Eswar et al., 2006*), was prepared with an automated pipeline for docking (*Irwin et al., 2009*) (*Supplementary files 5* and *6*). The proposed active site for each enzyme was identified by

superimposing liganded structures of closely related family members or related domains; for the enzymes considered here, identical results would have been obtained by identifying the largest cavity on the structure, for example, by using program PocketPicker (*Coleman and Sharp, 2010*). Cofactors (as generated by PRODRG server [*Schüttelkopf and van Aalten, 2004*]), metal ions, and water molecules were included in the protein structure preparation where required for enzyme function (*Irwin et al., 2005*; *Korczynska et al., 2014*; *London et al., 2014*). The KEGG database (*Kanehisa et al., 2016*; *Kanehisa and Goto, 2000*) of 14,212 unique metabolites from the ZINC database (*Irwin et al., 2012*) was docked against each target with DOCK3.6 in an automated fashion (*Mysinger and Shoichet, 2010*) (http://dock.compbio.ucsf.edu/). Compounds were ranked by a physics-based scoring function that evaluates ligand-protein complementarity considering van der Waals and electrostatic interactions, corrected for ligand desolvation (*Mysinger and Shoichet, 2010*; *Meng et al., 1992*; *Wei et al., 2002*). For each protein, the docking scores were converted to Z-scores by subtracting the mean and dividing by the standard deviation of the docking scores. The docking Z-score for the entire pathway model is the normalized sum of the Z-scores for all substrate-enzyme and product-enzyme pairs:

$$Z_{VS} = -\frac{1}{N}\sum_i^N Z_i,$$

where $N$ is the total number of enzyme-substrate and enzyme-product pairs in the pathway model. Similar normalizations of docking scores have been described for other applications (*Casey et al., 2009*).

## Chemical transformations

Chemical transformations derived from protein family annotations can help identify the substrate and product of an enzyme. These generic chemical transformations describe the enzymatic reaction without precise knowledge of the substrate, encoding the differences between the reactant and product on a more general level. While the full Enzyme Commission (EC) number describes the substrate specificity of an enzyme, the generic chemical transformation typically corresponds to the third level EC classification (*Hatzimanikatis et al., 2005*). For example, a serine acetyltransferase is a transferase that catalyzes the reaction of converting an alcohol into an ester. Pathway models with substrate-product pairs that match the chemical transformations are more likely to be correct than those pairs that do not (however, the final predicted reactions reflect the totality of all restraints, not only chemical transformation restraints).

The chemical transformation is determined from the Pfam classification (*Finn et al., 2016*) from the generic chemical reaction or reactions that are conserved across members of the protein family. Multiple chemical transformations may be considered for a single family. Relying on the library generation tool in OEChem Tools, the in silico transformation using SMIRKS strings was performed on each metabolite, represented as a SMILES string (*Supplementary files 5* and *6*). SMIRKS strings encode a generic reaction composed of a structural motif or pattern in the substrate and the corresponding pattern in the resulting product. Using RDKit, the Tanimoto coefficient between the transformed molecule and every other metabolite is computed based on the Morgan fingerprints, which are graph-based circular fingerprints useful for structural comparisons, with chirality taken into account. Because no transformation is defined for a dummy node, the Tanimoto coefficient between the substrate of the dummy node and every other metabolite is computed. For molecules with undefined stereocenters, the highest Tanimoto coefficient for up to 16 distinct stereoisomers was used.

Tanimoto coefficients were converted into Z-scores, similarly to the docking Z-scores. The score for transformations is the average Z-score over the Z-scores of all substrate-product-enzyme node triads in a model:

$$Z_{CT} = \frac{1}{N}\sum_i^N Z_i,$$

where $N$ is the total number of substrate-product-enzyme triads in the pathway model.

## Similarity ensemble approach

Comparison of ensembles of ligands using the Similarity Ensemble Approach (SEA) version 1.0 can predict functionally-linked proteins from the similarity of their ligands (*Keiser et al., 2007*), irrespective of their sequence or structural similarities (*Lin et al., 2013*). It is more likely that enzymes with high ligand similarity than enzymes with low ligand similarity are adjacent to each other in a pathway, as exemplified by a DUDE analysis (*Irwin et al., 2012*) (*Figure 2—figure supplement 1C*). Ensembles of predicted ligands can be obtained from virtual screening and used to restrain the identities of pairs of enzymes in a pathway (*Fan et al., 2013*). The top 500 docking-ranked metabolites for each enzyme were considered as the ligand ensemble. SEA E-values were calculated based on the similarity between these top 500 metabolites for each pair of enzymes in a putative pathway. The SEA E-value reflects the significance of the similarity between ligand ensembles for a pair of enzymes, compared to an expectation for two similarly sized sets of randomly selected metabolites from KEGG.

Specifically, the SEA E-value ($e_{value}$) for two consecutive pathway proteins $A$ and $B$ is first converted into the $S_{AB}$ score:

$$S_{AB} = w_{AB} * F_{AB} , \quad w_{AB} = \min(-\log(e_{value}), \, 50)$$

where $F_{AB}$ is $\frac{1}{50}\min(w_{AA}, w_{BB})$, modeling our confidence in the SEA analysis. Next, the $S_{AB}$ score is normalized into a Z-score $S_i$ by subtracting the mean and dividing by the standard deviation obtained from the distribution of $S_{AB}$ scores for all pairs of input enzymes, whether or not they are linked in the pathway. Finally, the SEA component ($Z_{SEA}$) of the integrative pathway score $Z_{Pathway}$ is the $S_i$ score averaged over all consecutive protein pairs in the tested pathway:

$$Z_{SEA} = \frac{1}{N}\sum_{i}^{N} S_i$$

## Pathway nodes and boundaries

Any known nodes of the pathway can be easily specified as constraints on the search for pathways that satisfy all input information. In the particular case of the L-gulonate catabolic pathway, a solute-binding protein (SBP) subunit of a TRAP transporter was identified, based on its strong sequence similarity to the TRAP SBP family. Thus, this transporter defines the start of the metabolic pathway to be modeled, with the rest of the pathway corresponding to intracellular enzymes acting on the transporter's substrate in series.

Similarly, knowledge about the endpoints of metabolic pathways can also constrain integrative pathway mapping. For catabolic pathways of sugars, we assume that the pathway produces a compound that feeds into central carbohydrate metabolism. Therefore, pathways in which the final product is a compound in central metabolism are more likely to be correct (*Supplementary file 4*).

The final metabolite in the model is compared by the Tanimoto coefficient using RDKit Morgan Fingerprints to all metabolites in central metabolism, and the maximum Tanimoto coefficient is used. The central metabolism endpoint score is:

$$Z_{CM} = \frac{TC - \overline{TC}}{SD},$$

where $TC$ is the Tanimoto coefficient between the final product in the pathway model and the most similar compound in central metabolism. $\overline{TC}$ is the average and $SD$ is the standard deviation of Tanimoto coefficients for all comparisons between each candidate metabolite and all compounds in central metabolism.

## Differential scanning fluorimetry screening hits

In our predicted pathway, L-gulonate and D-mannonate were identified as hits for the TRAP solute binding protein (ie, the first protein in the pathway), using screening of 189 compounds by DSF (*Vetting et al., 2015*). We assume that true substrates have high chemical similarity to the screening hits. Thus, the hits are compared by the Tanimoto coefficient using RDKit Morgan Fingerprints to the substrate of the screened enzyme in the pathway model.

$$Z_{HTS} = \frac{TC - \bar{TC}}{SD},$$

where $TC$ is the Tanimoto coefficient between the substrate in the pathway model and the hit in the screening assay. For multiple screening hits, the maximum of the Tanimoto coefficients between the substrate and hits is used. $\bar{TC}$ is the average and $SD$ is the standard deviation of Tanimoto coefficients between the substrate and all metabolites.

## Homology to a characterized enzyme

Substrates of enzymes in the pathway model are expected to be the same or similar to substrates of homologs that share high sequence similarity. For the L-gulonate pathway, the dehydratase has 73% sequence identity to a characterized mannonate dehydratase in *E. coli*. Therefore, pathways in which the substrate of the dehydratase is similar to D-mannonate are more likely to be correct than others. The proposed dehydratase substrate in an evaluated pathway model was compared to D-mannonate by the Tanimoto coefficient, using RDKit Morgan Fingerprints:

$$Z_{HS} = \frac{TC - \bar{TC}}{SD},$$

where $TC$ is the Tanimoto coefficient between the proposed substrate and the known substrate. $\bar{TC}$ is the average and $SD$ is the standard deviation of Tanimoto coefficients for all comparisons between each candidate metabolite and the known substrate.

## Stage 3: Sampling good models

With the scoring function in hand, the next step is to find pathway models that score well. These models are obtained by sampling candidate metabolites and proteins at each position in the linear pathway of a given length. We use Monte Carlo (MC) sampling by the Metropolis-Hastings algorithm with simulated annealing (*Hastings, 1970*; *Kirkpatrick et al., 1983*). The set of MC moves includes (i) swapping components of the same type within the graph and (ii) replacing a component in the graph with an unused candidate component of the same type. At each MC step, if the pathway score improves, the new model is accepted. Otherwise, the new model is accepted if a randomly sampled number from the uniform distribution between 0 and 1 is less than the acceptance probability computed by the standard Metropolis criterion:

$$p = \exp\left(\frac{-D}{T}\right)$$

where $D$ is the difference between the old and new pathway scores and $T$ is the simulated annealing 'temperature' parameter. The temperature drops over the course of the MC run:

$$T = 0.3 * 0.2^N + 0.1$$

where $N$ is the MC step number normalized by the total number of MC steps. With these parameters, a sampling run generally converges (*Figure 3—figure supplement 1E*), terminating after 5,000,000 MC steps. 1000 independent runs are performed, and the models sampled from all runs are combined. The unique sampled models with a score above a cutoff (good-scoring models) are considered in the analysis; the cutoff is two standard deviations below the best score. For the glycolysis pathway, which is about twice as long as the other pathways, a more stringent cutoff of 1.5 standard deviations was used such that the number of good-scoring pathways is comparable to that for the other benchmark pathways. The standard deviation is calculated from a distribution of scores of random models. Convergence of sampling was tested by determining the fraction of unique clusters as a function of the number of independent runs (*Figure 3—figure supplement 1*).

## Stage 4: Analyzing models and information

The resulting ensemble of good-scoring models is analyzed in terms of its precision and satisfaction of the restraints (*Figure 2*). Using hierarchical clustering in the scikit-learn python package

(**Pedregosa et al., 2011**), the pathway models are clustered by a pairwise distance metric (here, the Hamming distance). The Hamming distance is the number of positions at which the corresponding nodes are differently divided by the number of nodes in the pathway (including both proteins and ligands). Clusters are determined at the cutoff distance of 0.2.

A Cytoscape (**Shannon et al., 2003**) app, NetIMP, was written to perform visualization of the models (**Figure 1—figure supplement 3**). The app takes as input a JSON formatted file of with the IMP results, including the overall scores and the scores for the individual restraints. The app constructs a 'union network' that is the union of all models and presents it to the user. A slider is available to adjust the minimum score for inclusion in the union network. Each of the corresponding models are shown in a results panel to the right of the network. The user can highlight the model in the relevant network by selecting the row. Checkboxes allow the user to view the individual restraints, including whether or not the model satisfies the restraint. NetIMP is available in the Cytoscape app store.

## Computational cost

The total computation time depends on the numbers of enzymes and ligands considered. The computationally most demanding part of integrative pathway mapping is the preparation of the input information, not the sampling of the alternative pathways. For the three benchmark and L-gulonate pathways, the computing times for various steps in the process are as follows. Parallelized virtual screening of a library of ~20,000 compounds by docking each ligand against each of ~10 enzymes takes just over an hour on a cluster of 1400 nodes. The SEA analysis takes ~15 min to screen ~20,000 compounds against ~10 target proteins on 10 computing nodes. Finally, chemical transformation calculations take ~1 hr for 10 enzymes on a single node. Once the inputs are in hand, the runtime for a single Monte Carlo sampling run is approximately 1 hr on a single computing node; a few hundred independent Monte Carlo optimizations are typically performed in parallel on a cluster of compute nodes. In conclusion, the entire process, from preparation of inputs to the sampling of the pathways, can be performed in a few hours on a cluster of a few hundred nodes.

## Benchmarking

We assess our method on three known metabolic pathways, including the glycolysis pathway (10 enzymes, 2965 potential ligands), CMP KDO-8P biosynthesis pathway (four enzymes, 3336 potential ligands), and serine biosynthesis (five enzymes, 3494 potential ligands) (**Supplementary file 1**). Pathway docking was performed against crystal structures and comparative models (**Supplementary file 5**). The score of the pathway model is a sum of the individual Z-score terms for the docking screen, SEA calculation, and chemical transformations. The ability to recover the true ligand-enzyme pair is evaluated by the relative frequency of the ligand-enzyme edge occurring in the good-scoring pathway models. Therefore, we compare the rank of the substrate or product for a given enzyme based on the initial docking score to the rank based on the frequency of the corresponding ligand-enzyme edge.

## Benchmark assessment for decoy and dummy enzymes

The enzyme composition of a pathway is not always known with certainty, so we considered other scenarios beyond the simple case (**Figure 3AB**). We tested whether or not our method could detect the correct pathway when non-pathway enzymes, or decoys, are included in the initial candidate set of enzymes. An additional term that considers membership in the same gene cluster is included in the scoring function. Conserved gene clusters identified from comparative genome neighborhood analysis (**Figure 1—figure supplement 2**) can be informative about the functional relationships of genes acting in the same pathway (**Overbeek et al., 1999**). Pathway models with all members of a gene cluster are more likely to be correct than those that are missing members or containing non-members.

The set of protein pairs associated with the gene cluster identified by genome neighborhood analysis is compared to sets of protein pairs from all possible subsets of the protein candidates. The intersection between the sets of pairs is normalized by the larger of the number of pairs in the sets. The possible subsets range in number from as few as three to the number of total candidate proteins. The gene cluster score is:

$$Z_{GC} = \frac{GC - \bar{GC}}{SD},$$

where $GC$ is the normalized intersection between the set of protein pairs in the gene cluster and the set of protein pairs in the pathway model. $\bar{GC}$ is the average and $SD$ is the standard deviation of the normalized intersection for all comparisons between the proteins associated with the gene cluster and possible subsets of candidate proteins.

For the case in which the candidate set of enzymes is incomplete, a dummy enzyme that represents an unknown pathway enzyme was used (*Figure 2—figure supplement 1D*). For each of the pathways, one enzyme was replaced with a dummy enzyme in the initial set of candidate enzymes. For the serine biosynthesis case, the correct pathway with the dummy enzyme included was ranked as the top-scoring model (*Figure 2—figure supplement 1D*). For the other test cases, the inclusion of the dummy lowered the overall ranking of the correct pathway model, but the correct pathway was still within the top-scoring models.

## Experimental methods

### Cloning, expression, and purification of *Hi*UxuB, *Hi*KdgK, and *Hi*KdgA

The genes *HiUxuB* (Uniprot ID P44481), *HiKdgK* (Uniprot ID P44482), and *HiKdgA* (Uniprot ID P44480) were amplified from *H. influenzae* strain Rd KW20 (ATCC 51907) genomic DNA. PCR was performed using KOD Extreme DNA Polymerase (Novagen) according to the manufacturer's guidelines. The conditions were: 2 min at 95°C, followed by 40 cycles of 20 s at 95°C, 20 s at 66°C, and 20 s at 72°C. Primers are listed in *Supplementary file 7*. The amplified fragments were cloned into the N-terminal TEV cleavable 6x-Histag containing vector pNIC28-Bsa4 (pSGC-His), by ligation-independent cloning (*Aslanidis and de Jong, 1990*; *Savitsky et al., 2010*).

The *HiUxuB*-pSGC-His, *HiKdgK*-pSGC-His, *HiKdgA*-pSGC-His vectors were transformed into BL21 (DE3) *E. coli* containing the pRIL plasmid (Stratagene) and used to inoculate a 20 mL 2 x YT culture containing 25 µg/mL Kanamycin or 100 µg/mL Carbomycin and 34 µg/mL Chloramphenicol. The cultures were grown overnight at 37°C in a shaking incubator. The overnight culture was used to inoculate 2 L of PASM-5052 auto-induction media containing 150 mM 2–2-bipyridyl, 1 mM $ZnCl_2$, and 1 mM $MnCl_2$ (*Studier, 2005*) that was incubated at 37°C in a LEX48 airlift fermenter for 4 hr and then at 22°C overnight. The culture was harvested and pelleted by centrifugation.

Cells were suspended in lysis buffer (20 mM HEPES, pH 7.5, 500 mM NaCl, 20 mM imidazole, and 10% Glycerol) and lysed by sonication. The lysate was clarified by centrifugation at 35,000 x g for 30 min, loaded onto a 5-mL Strep-Tactin column (IBA) on an AKTAxpress FPLC (GE Healthcare), and washed with five column volumes of lysis buffer, and eluted in StrepB buffer (20 mM HEPES, pH 7.5, 500 mM NaCl, 20 mM Imidazole, 10% glycerol, and 2.5 mM desthiobiotin). The eluent was loaded onto a 1-mL HisTrap FF column (GE Healthcare), washed with 10 column volumes of lysis buffer, and eluted in buffer containing 20 mM HEPES pH 7.5, 500 mM NaCl, 500 mM Imidazole, and 10% glycerol. The purified sample was loaded onto a HiLoad S200 16/60 PR gel filtration column, which was equilibrated with SECB buffer (20 mM HEPES, pH 7.5, 150 mM NaCl, 10% glycerol, and 5 mM DTT). Peak fractions were collected, protein was analyzed by SDS-PAGE, concentrated to 2.4 g/L, 1.9 g/L, and 2.0 g/L, respectively, flash frozen in liquid nitrogen, and stored at −80°C.

### Cloning, expression, and purification of *Hi*GulD and *Hi*UxuA

The genes *HiGulD* (Uniprot ID Q57517) and *HiUxuA* (Uniprot ID P44488) were amplified from *H. influenzae* strain Rd KW20 (ATCC 51907) genomic DNA. PCR was performed using KOD Extreme DNA Polymerase (Novagen) according to the manufacturer's guidelines. The conditions were: 2 min at 95°C, followed by 40 cycles of 20 s at 95°C, 20 s at 66°C, and 20 s at 72°C. Primers are listed in *Supplementary file 7*. The amplified fragment was cloned into the C-terminal TEV cleavable 10x-Histag containing vector pNYCOMPS-LIC-TH10-ccdB (C-term) such that the tag is out of frame (pNYCOMPSC-tagless), by ligation-independent cloning (*Aslanidis and de Jong, 1990*; *Savitsky et al., 2010*).

The pNYCOMPSC-tagless *HiGulD* and *HiUxuA* constructs were transformed into *E. coli* BL21 (DE3) for expression. Both *Hi*GulD and *Hi*UxuA were purified from 1 L of culture using DEAE

Sepharose, Q-Sepharose, and phenyl-Sepharose columns (all Amersham Biosciences) as previously described (*Wichelecki et al., 2014*). Proteins were concentrated to 15 g/L and 6 g/L, respectively, flash frozen in liquid nitrogen, and stored at −80°C.

## Preparation of 2-keto-3-deoxy-D-gluconate

2-Keto-3-deoxy-D-gluconate was synthesized via an enzymatic procedure. The reaction (1.5 mL) contained 50 mM potassium HEPES, pH 7.9, 10 mM $MgCl_2$, 100 mM D-mannonate, and 1 µM D-mannonate dehydratase (Uniprot ID B0T0B1). The reaction was left to proceed at 37°C for 48 hr. Afterward, the enzyme was removed by filtration using 30,000 NMWL ultrafiltration membranes (Millipore). The identity of the product was verified via $^1$H-NMR.

## Preparation of 2-keto-3-deoxy-D-gluconate-6-phosphate

2-Keto-3-deoxy-D-gluconate-6P was synthesized via an enzymatic procedure. The reaction (1.5 mL) contained 100 mM potassium HEPES, pH 7.9, 10 mM $MgCl_2$, 120 mM ATP, 100 mM D-mannonate, 1 µM D-mannonate dehydratase (Uniprot ID B0T0B1), and 1 µM 2-keto-3-deoxy-D-gluconate kinase (Uniprot ID A4XF21). The reaction was left to proceed at 37°C for 48 hr. Afterward, the enzyme was removed by filtration using 10,000 NMWL ultrafiltration membranes (Millipore). The identity of the product was verified via $^1$H-NMR.

## Kinetic assays of L-gulonate catabolic pathway proteins

The kinetic assays were run in 200 µL aliquots at 37°C and monitored using a continuous spectrophotometric assay. The identities of all products were verified via $^1$H-NMR.

Oxidation was quantitated by measuring the increase in absorbance at 340 nm ($\varepsilon = 6220$ $M^{-1}$ $cm^{-1}$) of L-gulonate at carbon-5 by *Hi*GulD (50 mM Tris, pH 9, 1.5 mM $NAD^+$, and 200 nM *Hi*GulD). The substrate concentration was varied from 100 µM to 10 mM.

Oxidation was quantitated by measuring the increase in absorbance at 340 nm ($\varepsilon = 6220$ $M^{-1}$ $cm^{-1}$) of D-mannonate at carbon-5 by *Hi*UxuB (50 mM Tris, pH 9, 10 mM $MgCl_2$, 1.5 mM $NAD^+$, and 2 nM *Hi*UxuB). The substrate concentration was varied from 50 µM to 5 mM.

Dehydration was quantitated by measuring the decrease in absorbance at 340 nm ($\varepsilon = 6220$ $M^{-1}$ $cm^{-1}$) of D-mannonate by *Hi*UxuA (50 mM potassium HEPES, pH 7.9, 10 mM $MgCl_2$, 1.5 mM ATP, 1.5 mM PEP, 0.16 mM NADH, 9 units of pyruvate kinase, 9 units of lactate dehydrogenase, 18 units of 2-keto-3-deoxy-D-gluconate kinase, and 200 nM *Hi*UxuA). The substrate concentration was varied from 100 µM to 30 mM.

Phosphorylation was quantitated by measuring the decrease in absorbance at 340 nm ($\varepsilon = 6220$ $M^{-1}$ $cm^{-1}$) of 2-keto-3-deoxy-D-gluconate by *Hi*KdgK in a coupled assay with lactate dehydrogenase (50 mM potassium HEPES, pH 7.9, 10 mM $MgCl_2$, 1.5 mM ATP, 1.5 mM PEP, 0.16 mM NADH, 9 units of pyruvate kinase, 9 units of lactate dehydrogenase, and 200 nM *Hi*KdgK). The substrate concentration was varied from 100 µM to 5 mM.

Cleavage was quantitated by measuring the decrease in absorbance at 340 nm ($\varepsilon = 6220$ $M^{-1}$ $cm^{-1}$) of 2-keto-3-deoxy-D-gluconate-6P by *Hi*KdgA in a coupled assay with lactate dehydrogenase (50 mM potassium HEPES, pH 7.9, 10 mM $MgCl_2$, 1.5 mM PEP, 0.16 mM NADH, 9 units of lactate dehydrogenase, and 200 nM *Hi*KdgA). The substrate concentration was varied from 100 µM to 5 mM.

## Bacterial strains and growth conditions

*H. influenzae* Rd KW20 (ATCC 51907) was grown aerobically at 37°C with shaking at 225 rpm, and was routinely cultured in Brain Heart Infusion (BHI, Difco) broth or on BHI solid medium, supplemented with nicotinamide adenine dinucleotide (NAD) and hemin at 10 µg $mL^{-1}$ (sBHI). For gene expression analyses and carbon utilization studies, the defined medium of Coleman et al. (2003) was used. Glucose-free RPMI-1640 (Sigma R1383) was supplemented with the following additives: HEPES, 6 mg $mL^{-1}$; $NaHCO_3$, 2 mg $mL^{-1}$; inosine, 1.75 mg $mL^{-1}$; uracil, 87 µg $mL^{-1}$; NAD, 10 µg $mL^{-1}$; hemin, 10 µg $mL^{-1}$. D-glucose, L-gulonate, or D-mannonate (10 mM) served as the source of carbon. Kanamycin was added at 10 µg $mL^{-1}$ when appropriate.

## Growth curves

Growth curves were recorded using the Bioscreen C instrument (Growth Curves, USA) and 100-well plates. Starter cultures of *H. influenzae* Rd KW20 were grown overnight in sBHI, washed in minimal medium lacking carbon source, and re-suspended in an equivalent volume of minimal medium lacking carbon source. Each well contained 300 μL minimal medium with D-glucose, L-gulonate, or D-mannonate (10 mM), and was inoculated to 1% with washed starter culture. Plates were incubated at 37°C with continuous shaking at medium amplitude and the optical density at 600 nanometers (OD$_{600}$) was recorded every 30 min for 48 hr.

## Transcriptional analysis

Starter cultures of *H. influenzae* Rd KW20 were grown overnight in sBHI, washed in minimal medium lacking carbon source, and re-suspended in an equivalent volume of minimal medium lacking carbon source. This culture was used to inoculate 5 mL minimal medium (1% inoculum) with 10 mM glucose, and cultures were grown until OD$_{600}$0.3–0.5. Cells were washed and re-suspended in 4 mL minimal medium lacking carbon source. Cultures were divided into two equal 2 mL volumes, 10 mM glucose was added to one volume and 10 mM L-gulonate or D-mannonate was added to the other, and the cultures were grown until OD$_{600}$0.8–1.0. At the time of cell harvest, one volume of RNAprotect Bacteria Reagent (Qiagen) was immediately added to two volumes of each actively growing culture. Samples were mixed by vortexing for 10 s and incubated for 5 min at room temperature. Cells were pelleted, flash frozen in liquid nitrogen, and stored at −80°C.

RNA isolation was performed in an RNAse-free environment at room temperature using the RNeasy Mini Kit (Qiagen) according to the manufacturer's instructions. Cells were disrupted according to the 'Enzymatic Lysis Protocol' in the RNAprotect Bacteria Reagent Handbook (Qiagen); lysozyme (Thermo-Pierce) was used at 15 mg/mL. RNA concentrations were determined by absorption at 260 nanometers (A$_{260}$) using the Nanodrop 2000 (Thermo) and absorption ratios A$_{260}$/A$_{280}$ and A$_{260}$/A$_{230}$ were used to assess sample integrity and purity. Isolated RNA was stored at −80°C until further use.

cDNA synthesis was performed using 300 ng of total isolated RNA with the ProtoScript First Strand cDNA Synthesis Kit (NEB), according to the manufacturer's instructions. Primers for quantitative real-time (qRT) PCR were designed using the Primer3 primer tool; amplicons were 150–200 bps in length. Primers were 18 to 24 nucleotides in length and had a theoretical T$_m$ of 55–60°C. Primer efficiency was determined to be at least 90% for each primer pair. Primer sequences are provided in *Supplementary file 8*. qRT-PCRs were carried out in 96-well plates using the LightCycler 480 II instrument (Roche) with the LightCycler 480 SYBR Green I Master Mix (Roche), according to the manufacturer's instructions. Minus-RT controls were performed to verify the absence of genomic DNA in each RNA sample, for each gene target analyzed. Relative changes in gene expression were analyzed by the $2^{-\Delta\Delta CT}$ method (*Livak and Schmittgen, 2001*), using the 16S rRNA gene as a reference. Each qRT-PCR was performed in triplicate, and fold-changes are the averages of at least three biological replicates.

## Gene disruption

To create a genetic deletion of the putative L-gulonate SBP (HI0052), triple overlap extension PCR was used. Briefly, using Pfu Ultra High-Fidelity DNA polymerase (Agilent), three PCR fragments were generated: (a) a fragment corresponding to the genomic region ~1000 bps upstream of HI0052 was amplified from *H. influenzae* Rd KW20 genomic DNA with primers Del_HI0052_arm1fwd and Del_HI0052_arm1rev, (b) the kanamycin resistance cassette from p34s-Km (*Dennis and Zylstra, 1998*) was amplified with primers Kan_OL_delHI0052_fwd and Kan_OL_delHI0052_rev, and (c) a fragment corresponding to the genomic region ~1000 bps downstream of HI0052 was amplified from *H. influenzae* Rd KW20 genomic DNA with primers Del_HI0052_arm2fwd and Del_HI0052_arm2rev. The 3' end of fragment 'a' and the 5' end of the kanamycin resistance cassette (fragment 'b') were engineered with 50 bps of identical overlapping sequence, as were the 3' of the kanamycin resistance cassette and the 5' end of fragment 'c'. One hundred ng of each of these PCR fragments were combined in a triple overlap extension PCR with primers Del_HI0052_arm1fwd and Del_HI0052_arm2rev to generate a ~3 kb fragment with arms homologous to the genomic regions flanking HI0052, with an intervening kanamycin resistance cassette.

The same approach was used to create a genetic deletion of the putative L-gulonate dehydrogenase (HI0053). To generate the triple overlap extension product for deletion of HI0053, primers Del_HI0053_arm1fwd and Del_HI0053_arm1rev and primers Del_HI0053_arm2fwd and Del_HI0053_arm2rev were used to amplify the regions ~1000 bps upstream and downstream of HI0053, respectively, from *H. influenzae* Rd KW20 genomic DNA. The kanamycin resistance cassette from p34s-Km was amplified with primers Kan_OL_delHI0053_fwd and Kan_OL_delHI0053_rev. One hundred ng of each of these PCR fragments were combined in a triple overlap extension PCR with primers Del_HI0053_arm1fwd and Del_HI0053_arm2rev to generate a ~3 kb fragment with arms homologous to the genomic regions flanking HI0053, with an intervening kanamycin resistance cassette. Primer sequences are provided in *Supplementary file 9*.

Each of the triple overlap PCR products was gel-purified and 100 ng was transformed into 1 mL of *H. influenzae* Rd KW20 cells made competent by the M-IV method (*Poje and Redfield, 2003*). Double crossover recombinants were selected by resistance to kanamycin and confirmed by genomic PCRs.

## Cell preparation and metabolite extraction

GC-MS-based metabolic analysis of whole cell extracts was carried out with samples of *H. influenzae* Rd KW20 grown with $^{13}$C labeled L-gulonate or D-glucose, following the procedure of *Zhao et al. (2013)*. Cells grown in rich medium were diluted 1:100 into defined medium with 10 mM unlabeled L-gulonate or D-glucose as added carbon source and grown to an $OD_{600}$ of 0.6 (approximately 18 hr). Cells were harvested by centrifugation (4000 $\times$ g, 10 min, 4°C), washed twice in defined medium without added carbon source, and re-suspended in this medium. Cell density was adjusted to $OD_{600}$ = 6.0, and the cell suspension was then depleted of catabolic metabolites by incubation at 37°C for 30 min before transferring to ice. A mixture of 5 mM $^{13}$C labeled L-gulonate plus 5 mM unlabeled L-gulonate, or a mixture of 5 mM $^{13}$C labeled D-glucose plus 5 mM unlabeled D-glucose, was added to the samples followed by incubation at 37°C. At time points of 1, 2, 10, and 60 min, samples were pelleted by centrifugation (16,000 $\times$ g for 1 min), supernatants were removed, and cell pellets were flash frozen in liquid nitrogen. Samples were stored at $-80$°C prior to extraction. Metabolites were extracted directly from cell pellets by re-suspension in 0.5 mL extraction buffer (40:40:20 mixture of methanol:acetonitrile:water spiked with 1 mM L-norvaline for an internal standard) followed by 10 min of vortexing at room temperature. Cell extracts were cleared of debris via two rounds of centrifugation at 16,000 $\times$ g for 1 min, split into two equal portions, dried, and stored at $-80$°C prior to analysis.

Cell extracts were derivatized by one of two methods. To determine labeling of small metabolites of central carbon metabolism (glycolysis, TCA cycle and amino acids), extracts were derivatized with isobutylhydroxylamine and N-tert-butyldimethylsilyl-N- methyltrifluoroacetamide (TBDMS), and analyzed by GC-MS as described before (*Ratnikov et al., 2015*). Data were calculated in terms of fractional $^{13}$C labeling (the average $^{13}$C labeling across all metabolite carbons). Alternatively, to determine the labeling of gulonate and other 6-carbon molecules in the proposed pathway for L-gulonate metabolism, extracts were derivatized with 30 μL methoxyamine hydrochloride (Sigma, 20 mg/ml in pyridine) for 20 min at 80°C, followed by 30 μL BSTFA (trimethylsilylating reagent, Thermo) for 60 min at 80°C. Derivatized metabolites were analyzed by GC-MS as described before (*Scott et al., 2011*), using a modified temperature gradient: initial temperature was 60°C, held for 4 min, rising at 20 °C/min to 280°C, held for 4 min. Metabolites were identified by matching elution times and mass fragment patterns to standards. Labeling data for D-fructuronate were calculated as the ratio of mass 268: mass 264. Mass 264 corresponds to the fragment of D-fructuronate containing the four carbon atoms C3-C6 of the carbon backbone plus the complete derivatized side-chains (methoxaminated keto group and three trimethylsilylated hydroxyl groups) of this C3-C6 fragment, formula $C_{14}H_{34}NO_4Si_3$. Mass 268 corresponds to the same fragment with the backbone carbons $^{13}$C-labeled.

## Code availability

The source code for the IMP program, benchmark, input scripts files, and output files for the benchmark and the gulonate pathway calculations are available at http://integrativemodeling.org and https://github.com/salilab/pathway_mapping (*Calhoun, 2017*; copy available at https://github.com/

elifesciences-publications/pathway_mapping). Open access metabolite docking and libraries at http://metabolite.docking.org/ and at http://blaster.docking.org/. NetIMP is available in the Cytoscape app store.

## Acknowledgements

We thank Robert Munson for helpful discussions and advice on microbiological methods for *Haemophilus influenzae*. Chakrapani Kalyanaraman generously provided docking data. Martin Steinegger helped explore alternative pathway sampling methods in the initial stages of development. We thank OpenEye Scientific software for the use of OEChem tools. This work was supported by NIH U54 GM093342 (to JAG) and by NIGMS P41-GM103311 to Dr Thomas E Ferrin for the development of computational tools and Resource for Biocomputing, Visualization, and Informatics (RBVI) at UCSF, which partially supports the work of Dr A Sali.

## Additional information

### Competing interests

Matthew P Jacobson: Consultant to and stockholder of Schrodinger LLC, which licenses, develops, and distributes some of the software used in this work. The other authors declare that no competing interests exist.

### Funding

| Funder | Grant reference number | Author |
|---|---|---|
| National Institutes of Health | U54 GM093342 | John A Gerlt |
| National Institute of General Medical Sciences | P41-GM103311 | Andrej Sali |

The funders had no role in study design, data collection and interpretation, or the decision to submit the work for publication.

### Author contributions

Sara Calhoun, Magdalena Korczynska, Conceptualization, Data curation, Software, Formal analysis, Validation, Investigation, Visualization, Methodology, Writing—original draft, Project administration, Writing—review and editing; Daniel J Wichelecki, Brian San Francisco, Formal analysis, Validation, Investigation, Writing—review and editing; Suwen Zhao, Validation, Investigation, Writing—original draft; Dmitry A Rodionov, David A Scott, John H Morris, Daniel Russel, Andrei L Osterman, Data curation; Matthew W Vetting, Henry Lin, Validation, Investigation; Nawar F Al-Obaidi, Resources, Supervision, Validation, Writing—review and editing; Matthew J O'Meara, Investigation; Steven C Almo, Resources, Funding acquisition, Writing—review and editing; John A Gerlt, Resources, Supervision, Funding acquisition, Project administration, Writing—review and editing; Matthew P Jacobson, Resources, Supervision, Funding acquisition, Writing—review and editing; Brian K Shoichet, Andrej Sali, Conceptualization, Resources, Data curation, Supervision, Funding acquisition, Writing—original draft, Writing—review and editing

### Author ORCIDs

Magdalena Korczynska http://orcid.org/0000-0002-1339-5553
Suwen Zhao http://orcid.org/0000-0001-5609-434X
Brian K Shoichet http://orcid.org/0000-0002-6098-7367
Andrej Sali http://orcid.org/0000-0003-0435-6197

### Decision letter and Author response

Decision letter https://doi.org/10.7554/eLife.31097.025
Author response https://doi.org/10.7554/eLife.31097.026

# Additional files

### Supplementary files

• Supplementary file 1. Benchmark pathways. Three well characterized pathways were used for benchmarking of the integrative pathway mapping: glycolysis (10 enzymes) (*Kalyanaraman and Jacobson, 2010*), cytidine monophosphate 3-deoxy-D-*manno*-octulosonate 8-phosphate (CMP KDO-8P) biosynthesis (four enzymes), and serine biosynthesis (five enzymes).
DOI: https://doi.org/10.7554/eLife.31097.014

• Supplementary file 2. Substrate and product ranks by enzyme using virtual screening and integrative pathway mapping for benchmark and prospective pathway construction. The number of candidate ligands as input is 3650 for the L-gulonate catabolic pathway. The ranks from individual docking runs are determined by docking score for an enzyme. The ranks by the integrative approach are determined from the relative frequency of ligand-enzyme pairs in the ensemble of high-scoring pathway models. This frequency is then used to order ligands for a given enzyme.
DOI: https://doi.org/10.7554/eLife.31097.015

• Supplementary file 3. DSF screening of a library of 189 compounds (*Vetting et al., 2015*) indicating ligands that stabilized thermal denaturation of SBPs. The $\Delta T$m for each ligand was calculated as the difference of the $T$m values measured with a ligand and without ligand for SBS from *Haemophilus influenzae* RdAW and *Mannheimia haemolytica* PHL213 (GulP: 82% sequence identity).
DOI: https://doi.org/10.7554/eLife.31097.016

• Supplementary file 4. Central carbon metabolism compound set.
DOI: https://doi.org/10.7554/eLife.31097.017

• Supplementary file 5. Structural models for virtual screening and chemical transformations of enzymes in benchmark pathways.
DOI: https://doi.org/10.7554/eLife.31097.018

• Supplementary file 6. Structural models for virtual screening and chemical transformations of proteins in L-gulonate catabolic pathway.
DOI: https://doi.org/10.7554/eLife.31097.019

• Supplementary file 7. Primer sequences used for cloning genes in L-gulonate catabolic pathway.
DOI: https://doi.org/10.7554/eLife.31097.020

• Supplementary file 8. qRT-PCR oligonucleotide sequences used for gene expression profiling.
DOI: https://doi.org/10.7554/eLife.31097.021

• Supplementary file 9. Genetic deletion mutants of the putative L-gulonate catabolism pathway in *H. influenzae* Rd KW20 were generated with the following oligonucleotide sequences.
DOI: https://doi.org/10.7554/eLife.31097.022

• Transparent reporting form
DOI: https://doi.org/10.7554/eLife.31097.023

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
