## [Decision Letter]

Thank you for submitting your article "Integrative mapping of enzymatic pathways" for consideration by *eLife*. Your article has been favorably evaluated by Michael Marletta (Senior Editor) and three reviewers, one of whom, Nir Ben-Tal (Reviewer #1), is a member of our Board of Reviewing Editors.

The reviewers have discussed the reviews with one another and the Reviewing Editor has drafted this decision to help you prepare a revised submission.

The manuscript introduces a novel approach to enzyme annotation based on analyzing the enzyme within the context of its metabolic pathway. Structural information is utilized to enhance systems models. In particular, predictions of the substrates and products of the enzymes, as well as structural data (from experiment and prediction) are used within the framework of Monte Carlo search to assemble putative metabolic pathways. The most highly ranked pathway is considered as hypothesis, and experiments are designed to examine it.

The strength of this study is that it is an intuitive, well-fleshed out method. Structural prediction data from docking, which is not so strong on its own for functional prediction, is coupled to locations within a pathway, restricting the search space of possible solutions to chemically similar compounds in the pathway proximity of each other. There is also strong experimental evidence for parts of the pathway prediction in *H. influenzae*.

The weakness of this study is that the initial starting points to build these pathways seems vague, pathway models are not fully experimentally verified (although parts have strong evidence), and application on a larger (genome) scale is not discussed, where this would be incredibly useful. An assumption is made to start with genome organization to select the pathway members which is unexplained and should be backed up.

Opinion:

It is an interesting approach but because of several hype statements the reader ends up somewhat disappointed. The approach is validated on three well known and one novel pathway. This is very impressive both in the scope of work and in the conceptual completeness, but potentially misleading as the computational approaches clearly depend on the amount and quality of already existing information. From experimental structures cocrystallized with correct ligands that would help in virtual screening, to reliance on well understood biochemistry and on homology to already characterized enzymes this approach seems to be aiming for semi-automated reanalysis of already well characterized pathways rather than on annotations of novel, uncharacterized ones – which seems to be the stated motivation for this work. In this context the benchmarks and the "discovery" of L-gulonate catabolic pathway in *H. influenzae* (which seems to be already well annotated in public databases) are not very convincing. It is also a bit unclear what constitutes actual results of this paper, as significant elements of L-gulonate catabolic pathway discovery in *H. influenzae* were already published by the same authors in 2015. The authors should either rebrand their approach as aiming at cleaning and solving apparent inconsistencies in existing annotations, or come up with more convincing examples of discovery and annotations of genuinely novel functions/pathways. Such examples would provide more realistic evaluation of the proposed approach. In summary: The main text should make it clearer as to what the immediate application is. Is this method currently limited to filling in parts of a pathway, or is it feasible to now try reconstructing entire cells from "bags" of proteins and metabolites? If the latter, suitable examples should be presented.

Major issues:

1) Assuming that the examples will not change, many statements should be tuned down, starting in the title and Abstract, to reflect the fact that: (a) the proposed approach can, at best, resolve conflicts in existing annotations rather than suggest new annotation, and (b) that even that requires a lot of manual interventions, as opposed to fully automated method.

2) The reasons for the data types chosen to be integrated over other potentially useful sources should be flashed. Reasons could include increasing the computational time, limitations of experimental data, etc.

3) Methods of picking the initial proteins should be explained, since that is a nebulous first step to get around before using this useful method.

4) Selecting based on proximity in genome organization is not backed up (subsection “Problem and approach”, last paragraph). Statistics or citations should be shown on known pathways if this is true. This is an important step to explain because it seems like it may ignore many enzymes which are not in the genome neighborhood. In eukaryotic organisms this does not seem to be nearly as applicable as to prokaryotes.

5) Computational time is unknown, throughout the paper – estimated time to run all docking predictions, chemical similarity, binding site locating, MC sampling, etc. What is the bottleneck? Or main source of human time/input?

6) Is docking done only to one predicted active site in an enzyme? How reliable or comprehensive the binding site estimates used here are? How much does docking add over geometric + biochemical comparisons of active sites? Would it be sufficient to do that and compute similarity analyses of that, rather than running many docking simulations?

7) Discussion, first paragraph. There is additional related work of genome-scale metabolic reconstructions that include structural data:

- http://journals.plos.org/ploscompbiol/article?id=10.1371/journal.pcbi.1000938 – structural properties used in predicting drug off target pathways.

- https://bmcsystbiol.biomedcentral.com/articles/10.1186/s12918-016-0271-6 – structural properties used to compare lifestyles and pathway usage.

8) Figure 4 – what is the reason behind higher gene expression increases of UxuB, GulPQM, GulD on a D-mannonate carbon source compared to L-gulonate? Since these proteins are involved in L-gulonate transport and metabolism it would seem like they should be expressed higher on L-gulonate carbon source, not D-mannonate which comes after them in the proposed pathway.

9) What is the author's opinion on applying this method on a genome scale (i.e. entire metabolic network)? Is it feasible in terms of speed and data curation? Is it possible for the authors to generate genome-scale network predictions and compare them to available reconstructed metabolic models? These would be interesting questions to address in the main text.

10) How broadly applicable the approach is? How many new enzymes can be annotated this way?

11) The statement concerning the applicability of the method also to protein-protein interaction (PPI) networks is a bit of a stretch. For now, the essence of the method is docking of the metabolites to their binding/catalytic sites. In PPI this component will have to be replaced, and it is unclear with what. Protein-protein docking, the first possibility that comes to mind, is far less accurate, especially for transient interactions. I would eliminate the statement, or tune is down significantly.

12) Methods:a) The terms in the first equation should be defined.

b) Using the DOCK scores. Usually, in drug discovery campaigns, the docking scores are considered an indication for possible binding, but the actual values or ranks are often inaccurate. It is surprising that here they are taken as a given and nevertheless the pipeline works fine.

---

## [Author Response]

Major issues:1) Assuming that the examples will not change, many statements should be tuned down, starting in the title and Abstract, to reflect the fact that: (a) the proposed approach can, at best, resolve conflicts in existing annotations rather than suggest new annotation, and (b) that even that requires a lot of manual interventions, as opposed to fully automated method.

We have edited the manuscript throughout to tone down overreaching statements. For example:

The title was changed to: **"**Prediction of enzymatic pathways by integrative pathway mapping”.

We rephrased claims that we discovered a new pathway, instead focusing on development of an approach to predict the functions of the individual enzymes. For example, in the Abstract, “A computational method that can identify” → “predicts”; and **“**Demonstrated the utility by discovering a L-gulonate pathway” → “predicted”. We also removed any mention of “discovery” of the pathway throughout the main text. For instance, “we sought to predict this putative pathway using integrative pathway mapping”.

The input is indeed not prepared entirely automatically, which is true for all automated methods as at some point inputs must be prepared at least in part “by hand”. However, once all the input information is in hand, our method does convert it into the output automatically. For example, the input for the L-gulonate pathway prediction was mainly docking results, chemoinformatics analysis, chemical transformations, genomic neighbourhood, and experimental data obtained as the project evolved. This information was then fed into integrative pathway mapping that automatically produced the output (i.e., the gulonate pathway).

We clarified this point: “The preparation of input information requires manual processing. […] Nevertheless, we emphasize that once the input information is provided, its conversion into the predicted pathway is automated and does not require human intervention.”

Also, we state: “While not all types of input for integrative pathway mapping can be obtained automatically (e.g., docking, experimental measurements), the mapping itself is entirely automated.”

As the reviewer expected, we have indeed not eliminated the description of the L-gulonate pathway from the manuscript. As described in the summary above, this was a genuinely new pathway prediction, one tested and supported in this work with an experimental rigor still relatively rare in the field.

2) The reasons for the data types chosen to be integrated over other potentially useful sources should be flashed. Reasons could include increasing the computational time, limitations of experimental data, etc.

A major goal of the manuscript is to introduce an integrative approach to pathway mapping. Accordingly, we selected several accessible types of input to demonstrate this idea, while suggesting that additional sources of information could be considered in the future. As more types of information are added to the integrative approach, they might further improve its accuracy, precision, and applicability. Docking, the chemoinformatic Similarity Ensemble Approach, and chemical reactions seemed like sensible initial inputs, as they exploit established methods. The application to the gulonate pathway illustrates the use of several other types of input, including experimental data from differential scanning fluorimetry (it can be done in high-throughput) and additional theoretical considerations (e.g., the end point of the gulonate pathway must be a known metabolite), thus illustrating the relative ease with which information about a pathway can be included.

In Materials and methods, we now add: “First, input information has to be collected from computational and/or experimental sources. […] Moreover, additional types of information can be added, hopefully improving the accuracy, precision, and applicability of the approach, as illustrated by the gulonate pathway prediction that also depends on differential scanning fluorimetry data, pathway anchor points, and protein homology considerations.”

3) Methods of picking the initial proteins should be explained, since that is a nebulous first step to get around before using this useful method.

The candidate enzymes involved in the gulonate pathway were selected by identifying genes that are conserved in the genome neighborhood of the TRAP SBP gene (which encodes the transporter known to import gulonate into the cell) across multiple bacterial species. This analysis involved the construction of a genome neighborhood network (Zhao et al., *eLife*, 2014), which displays conserved protein families found in proximity to the TRAP SBP gene.

Thus, we added the following description: “For example, for the gulonate pathway, we identified 5 metabolic enzymes that are conserved in the genome neighborhood of the TRAP transporter gene by constructing a genome neighborhood network (GNN); the GNN approach has been demonstrated to accurately predict enzymes and transporters that function together in metabolic pathways based on conserved protein families in genome neighborhoods across different species (Zhao et al., 2014).”

Also, specific details for the gulonate pathway are given in Materials and methods: “A sequence similarity network (SSN) and genome neighborhood network (GNN) were constructed using the EFI-EST webserver (Gerlt et al., 2015) and Pythoscape v1.0 software (Barber and Babbitt, 2012) for an anchor protein, TRAP SBPs (Uniprot ID P71336 and Uniprot ID A7JQX0), to provide candidate pathway members […] Analysis of the GNN identified five enzyme families as candidate pathway members, including two dehydrogenases, one sugar dehydratase, one carbohydrate kinase, and one aldolase. The genes associated with these families in *H. influenzae* are co-localized in the genome with the TRAP SBP gene.”

4) Selecting based on proximity in genome organization is not backed up (subsection “Problem and approach”, last paragraph). Statistics or citations should be shown on known pathways if this is true. This is an important step to explain because it seems like it may ignore many enzymes which are not in the genome neighborhood. In eukaryotic organisms this does not seem to be nearly as applicable as to prokaryotes.

We agree. Accordingly, we added a discussion of the relationship between the genome neighborhood and pathway membership for prokaryotes. We also point out that the relationship is not nearly as strong in eukaryotes, most likely invalidating this particular consideration for predicting eukaryotic pathways. We also note that the integrative framework allows for incorporating other information about pathway membership (e.g., based on cellular localization, homology, biochemical function, and direct experimental evidence), which might reduce the size of the input set of potential protein members.

In Discussion, we now write: “Moreover, no single type of input information is essential, provided sufficient information is available from other sources. […] Thus, the integrative approach is at least in principle not limited to prokaryotic pathways.”

Also, in section “Stage 2: Designing pathway model representation and evaluation” in Discussion, we added: “This step can be substituted or supplemented by any other method that identifies candidate genes, including but not limited to: 1) colocalization of genes providing operon/metabolic context for prokaryotic proteins (Overbeek et al., 1999), 2) coexpression measured through chip-based and RNA-seq technologies (Barber and Babbitt, 2012), 3) co-regulation predicted by upstream DNA motifs (Wang, Gerstein and Snyder, 2009; Pilpel, Sudarsanam and Church 2001), 4) protein-protein interaction studies (Rodionov, 2007; Bork et al., 2004), 5) protein fusion events (Meier, Sit and Quake, 2013), and 6) phylogenetic profiles across different genomes (Marcotte et al., 1999).”

5) Computational time is unknown, throughout the paper – estimated time to run all docking predictions, chemical similarity, binding site locating, MC sampling, etc. What is the bottleneck? Or main source of human time/input?

To address this point, we added a new section entitled “Computational cost”, in Methods): “The total computation time depends on the numbers of enzymes and ligands considered. […] In conclusion, the entire process, from preparation of inputs to the sampling of the pathways, can be performed in a few hours on a cluster of a few hundred nodes.”

6) Is docking done only to one predicted active site in an enzyme? How reliable or comprehensive the binding site estimates used here are? How much does docking add over geometric + biochemical comparisons of active sites? Would it be sufficient to do that and compute similarity analyses of that, rather than running many docking simulations?

For each enzyme, as we now describe in the section “Molecular docking screens”, the docking was run against a single predicted binding site. Identical predictions were obtained by two independent methods: (1) similarity to a protein structure with a known site and (2) identifying the largest cavity in the structure using the program PocketPicker (Irwin et al., 2009).

As the reviewer points out, one could take known ligands of proteins homologous to a potential pathway member and rank the corresponding protein ligand pairs higher than those with other potential ligands, perhaps considering the degree of similarity of active sites and/or ligands. We explored this idea, but did not incorporate it in the current version of the method because we do not always have information about homologous proteins and their ligands. Perhaps more importantly, benchmarking showed that virtual screening can potentially identify ligands more accurately than a simple homology-based ligand transfer (Fan et al., J. Chem. Inf. Model., 2009, PMID: 19845314) (now pointed out in the first paragraph of the Discussion). Nevertheless, homology-based transfer may be one of the scoring function terms to add in the future, to further improve our integrative approach.

7) Discussion, first paragraph. There is additional related work of genome-scale metabolic reconstructions that include structural data:- http://journals.plos.org/ploscompbiol/article?id=10.1371/journal.pcbi.1000938 – structural properties used in predicting drug off target pathways.- https://bmcsystbiol.biomedcentral.com/articles/10.1186/s12918-016-0271-6 – structural properties used to compare lifestyles and pathway usage.

We added these citations in the first paragraph of the Discussion.

8) Figure 4 – what is the reason behind higher gene expression increases of UxuB, GulPQM, GulD on a D-mannonate carbon source compared to L-gulonate? Since these proteins are involved in L-gulonate transport and metabolism it would seem like they should be expressed higher on L-gulonate carbon source, not D-mannonate which comes after them in the proposed pathway.

The relative natural abundances of the two carbon sources may impact the upregulation of genes involved for D-mannonate conversion. For example, D-mannonate is a part of the catabolic pathway of D-glucuronate, which is commonly found in nature. L-gulonate metabolism intersects with D-glucuronate metabolism at fructuronate, but we are unaware of any previous reports of the use of L-gulonate.

One explanation for the higher expression of *GulPQM* on a D-mannotate source is that the GulPQM transporter could have a lower binding efficiency for D-mannonate vs. L-gulonate (differential scanning fluorimetry results, Supplementary file 3). Furthermore, *UxuB* and *GulD* may be co-regulated with *GulPQM* and, thus, is also upregulated when grown on D-mannonate.

9) What is the author's opinion on applying this method on a genome scale (i.e. entire metabolic network)? Is it feasible in terms of speed and data curation? Is it possible for the authors to generate genome-scale network predictions and compare them to available reconstructed metabolic models? These would be interesting questions to address in the main text.

The framework of the approach can, in principle, take any information that can be used to rank alternative pathways, and then samples alternative pathways to find those that are highly ranked. Thus, it could take information used for constructing entire metabolic networks by others (or indeed other networks) and, given a sufficiently large computing facility and perhaps improved sampling algorithm, in principle predict entire metabolic networks. The quality of the output is limited by the amount of information and exhaustiveness of sampling. In practice, however, reconstructing entire metabolic networks is out of scope for this first paper and for the current implementation of the method. We do believe, as alluded to above, that the method can, even now, complement the genome reconstruction methods in the realm of “dark” metabolic space. Still, not wishing to mislead on this point and succumb to hype, we prefer to leave such possible future applications uncommented on in the manuscript. We beg your indulgence here.

We now end Discussion with an appropriately cautious statement: “With further development, the framework may be applicable on a larger scale, approaching complete genomes, but mapping topologies of networks will be more demanding as it will require more input information and larger computation.”

10) How broadly applicable the approach is? How many new enzymes can be annotated this way?

To answer this question to a degree, we consider the fraction of enzymes that have known X-ray structures or sufficiently high quality comparative protein structure models – the latter of which we use in this manuscript – for application of virtual screening: domains in about 70% of proteins have known or modelable structures (Pieper et al., Nucl Acids Res 42, 336-346, 2014); not every enzyme in the pathway needs to be analyzed by virtual screening (cf, the dummy enzyme benchmarking in Figure 2—figure supplement 1). Still, we estimate that not more than one half of known enzymatic pathways of comparable length to those benchmarked could be mapped by our method.

To test the generality of integrative pathway mapping approach, we are currently applying it to three prokaryotic pathways of interest to the Enzyme Function Initiative consortium, to which all authors belong. In particular, we are focusing our attention on predicting new reactions that have not been associated with a specific enzyme; all predictions are linked to a specific gene that can then be experimentally tested.

11) The statement concerning the applicability of the method also to protein-protein interaction (PPI) networks is a bit of a stretch. For now, the essence of the method is docking of the metabolites to their binding/catalytic sites. In PPI this component will have to be replaced, and it is unclear with what. Protein-protein docking, the first possibility that comes to mind, is far less accurate, especially for transient interactions. I would eliminate the statement, or tune is down significantly.

We agree, and have eliminated the statement.

12) Methods:a) The terms in the first equation should be defined.

We agree, and now define each of the terms in the equation.

b) Using the DOCK scores. Usually, in drug discovery campaigns, the docking scores are considered an indication for possible binding, but the actual values or ranks are often inaccurate. It is surprising that here they are taken as a given and nevertheless the pipeline works fine.

The reviewer correctly points out that the scores and the ranks in a docking campaigns are often inaccurate, which is why the integrative mapping approach relies on several inputs – and not only on one source of information such as docking – to reach a consensus ranking. However, docking can enrich for real ligands at the top of the docking list. Therefore, in the L-gulonate pathway example we limit the input to the top 500 docking-ranked metabolites. This is illustrated directly in Figure 2 where only a combination of all three scoring function terms results in a correct ordering of enzymes and substrate assignments, while scoring functions containing only two or less types of scoring terms result in imperfect predictions.